# A Destruction Model of the Vascular and Lymphatic Systems in the Emergence of Psychiatric Symptoms

**DOI:** 10.3390/biology10010034

**Published:** 2021-01-06

**Authors:** Kohei Segawa, Yukari Blumenthal, Yuki Yamawaki, Gen Ohtsuki

**Affiliations:** 1Department of Drug Discovery Medicine, Kyoto University Graduate School of Medicine, Kyoto 606-8397, Japan; widemeister@gmail.com (K.S.); yamawaki.yuki.87s@st.kyoto-u.ac.jp (Y.Y.); 2Urology Department at Cambridge University Hospitals, NHS Foundation Trust, Addenbrooke’s Hospital, Hills Road Cambridge, Cambridge CB2 0QQ, UK; yukari.blumenthal@doctors.org.uk

**Keywords:** meningeal lymph, lymphangiogenesis, glymphatic system, immune-cell infiltration into brain parenchyma, blood-brain barrier (BBB) breakdown, immune-triggered disruption of neuro-glial physiology, emergence of psychosis

## Abstract

**Simple Summary:**

Impairment of the immune-barrier system in the aged brain is manifested in neurodegenerative diseases, such as Alzheimer’s disease. Cerebrospinal fluid flow and drainage system of meningeal lymph scavenge the waste products in the brain parenchyma. Recent accumulating attention to the immune system in the brain is now accompanied by the expanding knowledge of the breakdown mechanism of the blood-brain barrier and increased permeability of activated immune cells at the glymphatic system, in developing brains, and in the situation of infections. These disruptions of the immune-barrier function between vasculature and brain parenchyma, where neurons locate, would also occur by infection of viruses or microorganisms, vascular injury, cerebral hemorrhage, neurological diseases, and other external factors like stress. Those activate microglia, brain-resident immune cells, and allow the infiltration of immune cells, such as macrophages, neutrophils, and T cells. Aberrant immunity disrupts the functions of neurons and prevents the maturation of glia (e.g., oligodendrocytes and astrocytes), which may cause the emergence of the highly variable psychiatric symptoms seen in schizophrenia, autism spectrum disorders, Alexander disease, and developmental disorders via an abnormality of brain activities at both cellular and organism levels. Therefore, this hypothesis challenges the longstanding neuron-centric view of neurodegenerative and psychiatric diseases.

**Abstract:**

The lymphatic system is important for antigen presentation and immune surveillance. The lymphatic system in the brain was originally introduced by Giovanni Mascagni in 1787, while the rediscovery of it by Jonathan Kipnis and Kari Kustaa Alitalo now opens the door for a new interpretation of neurological diseases and therapeutic applications. The glymphatic system for the exchanges of cerebrospinal fluid (CSF) and interstitial fluid (ISF) is associated with the blood-brain barrier (BBB), which is involved in the maintenance of immune privilege and homeostasis in the brain. Recent notions from studies of postmortem brains and clinical studies of neurodegenerative diseases, infection, and cerebral hemorrhage, implied that the breakdown of those barrier systems and infiltration of activated immune cells disrupt the function of both neurons and glia in the parenchyma (e.g., modulation of neurophysiological properties and maturation of myelination), which causes the abnormality in the functional connectivity of the entire brain network. Due to the vulnerability, such dysfunction may occur in developing brains as well as in senile or neurodegenerative diseases and may raise the risk of emergence of psychosis symptoms. Here, we introduce this hypothesis with a series of studies and cellular mechanisms.

## 1. Introduction

Lymph fluid conveys the interstitial fluid (ISF) containing liquid, electrolytes, blood gas for fluid homeostasis. Also, it helps to transport various antigens, activated antigen-presenting cells, and immune effector cells for immune surveillance. The lymphatic system has an independent circulation system that is distinct from blood vessels [1] (Figure 1). In a body, the lymphatic system is composed of a network of lymphatic vessels and lymph nodes. Essentially, lymph is a protective measure against bacterial and viral infections. However, the concept that the lymphatic organ exists in the brain had been neglected, and its functioning has not been investigated until recently [2,3,4,5]. While such lymphatic vasculature was originally recognized over 200 years ago by Giovanni Paolo Mascagni, who published “the Vasorum Lymphaticorum Corporis Humani Historia et Ichnographia” (Mascagni, 1787) [6,7]. Mascagni clearly described the meningeal lymphatic vessels and their physiological role, although it was difficult to address the physiological functions for neurons, glia, and brain activity at that time. It took more than a hundred years after him for it to be termed the immune privilege of the brain [7]. Now, magnetic resonance imaging (MRI) with a gadolinium-based contrast agent unveiled the existence of the meningeal lymphatic vessels in human and nonhuman primates (i.e., common marmoset monkeys) [8]. The lymphatic system in the brain has received a great deal of attention not only because it is associated with dementia and Alzheimer’s disease (AD) but also due to its association with psychiatric symptoms caused by various intracerebral immune disorders.

Our living body has evolutionally developed the innate and acquired immune systems to protect from diseases by recognizing and killing non-self-substances such as pathogens, foreign substances, and abnormal cells, such as cancer. The activation of immune cells, including leucocytes, modulates the activity of neurons and glial-cells like microglia, astrocytes, and oligodendrocytes in the central nervous system (CNS) [9,10,11]. The healthy brain capillaries hold specialized shields, and they can cleverly shut down the invasion of bacteria, viruses, and toxic agents as pathogens for immune surveillance via the blood-brain barrier (BBB) (Figure 2). The BBB *per se* selectively supplies substances required for brain maintenance from blood, and conversely, excretes unnecessary metabolic wastes into the blood. Now, it is recognized that, in a situation of aberrant immunity in many neurodegenerative and psychiatric diseases, both the breakdown of the BBB and immune-cell entry from the circulatory system to the brain parenchyma occur via the chemokine-signaling pathways [11,12,13,14,15]. While blood vessels are the circulation system that provides the tissues with the oxygen and nutrients required for their various functions, the lymphatic system keeps tissue homeostasis by recycling ISF and maintaining immune surveillance [1]. Therefore, the blood-lymph-circulation system in the CNS is not just an innate feeding system for the selection of flows of solutes. The brain’s lymphatic drainage is currently recognized as having functions to maintain the water and ion homeostasis of the ISF, clearance of waste products, and reabsorption of macromolecules in fluids. The lymphatic system of the brain has a physiological function for modulating immune surveillance and immune responses in the brain. Currently, the lymphatic system in the mouse and human brain were revealed [4,8,16]. The whole visualization of blood vessels by rendering brain tissue transparent [17] and a molecular atlas for the blood vascular and vessel-associated cells in mouse brains using vascular single-cell transcriptomics [18] are also available, and they are extremely helpful to expand our knowledge. Therefore, a lot of attention is paid to the fact that brain inflammation, lymphedema, aberrant immunity, hypertension, and obesity result in various alterations in the vascular and lymphatic systems, such as leakage, damage, cleavage, hypo- and hyper-plasia [19,20]. However, the link between the destruction of the immune system in the brain, especially in the course of development, to the emergence of psychiatric diseases is yet unsolved. Here, we attempt to integrate the recent findings into a conceptual framework that represents the current understanding of the roles of the immune cells and glia within the vascular and lymph systems in brain dysfunction.

## 2. Main Text

### 2.1. Vascular Lymphatic System in the Brain

#### 2.1.1. Brain Vasculature System

Until recently, a conventional lymphatic vasculature system had been unowned in the central nervous system (CNS). Here, we would like to stress one physician in history had discovered a structure of lymphatic vessels in the brain [6,7]. The present (re)discovery of the lymphatic network in the brain that drains waste products of tissues has impressed us with the significance of the drainage system in the brain for scavenging macromolecules and securing immune cells into the CNS ISF and CSF [2,4,21]. CSF is produced mainly by ependymal cells of the choroid plexus at the lateral ventricles and the cisterna magna: i.e., the posterior cerebellomedullary cistern, and they generate circulative movement of the fluids. Ependymal cells also have a fluid–brain barrier function between ventricles and parenchyma, which is called the blood-cerebrospinal fluid barrier (BCSFB). Ependymal cells form the choroid plexus, and have high water and ion permeability. The choroid plexus absorbs water and ions from the blood much more efficiently than the BBB and generates CSF in the ventricles and subarachnoid spaces (Figure 1). Arachnoid granulations, small protrusions of the arachnoid matter, allow CSF to pass from the subarachnoid space into the venous system (Figure 2). The circulation of ISF is driven by pulsation of arteries along the basement membrane and smooth muscles of the brain blood vessels, which unmyelinated nerves project. According to Kiviniemi et al. (2016) [22], other than cardiac pulsation (1.08Hz), they proved additional two types of physiological mechanisms affecting CSF/ISF pulsations: respiratory pulsation (0.37 Hz) and very low-frequency vasomotor pulsations (at 0.01–0.027 Hz and 0.027–0.073 Hz) [22]. Ciliary motility of ependymal cells is also known involved in the efficient and continuous maintenance of CSF flow near the ventricular surface [23]. While immune cells come to the brain via the blood vessels, meningeal lymphatic vessels are also capable of delivering immune cells to the brain. Louveau et al. (2015) showed lymphatic vessel endothelial hyaluronan receptor 1 (LYVE-1) positive abluminal vessels besides the dural sinuses, which contain the CD3e positive T cells [4]. LYVE-1 positive vessels run along the blood vessels covered by smooth muscles in the superior sagittal sinus of the meninges, as we discuss later. Thus, the brain lymphatic system has been considered to conduct extra-blood supply and its outflow [5]. Meningeal lymphatic vasculature assists in the drainage of CSF and allows immune cells to enter draining lymph nodes in a CC-chemokine receptor 7 (CCR7 (i.e., CD197))-dependent manner. Two chemokines (C-C motif) ligands, CCL19 and CCL21, have been identified for the receptor CCR7, which keeps the balance of immunity and tolerance in the body via regulation of naive and regulatory T cells and migration of dendritic cells (DCs) in the lymphoid organs [24,25,26].

#### 2.1.2. Characteristics of Meningeal Lymphatic System

In general, in the body, the lymph is first drained into the initial lymphatic vessels that have both lymphatic capillaries, composed of lymphatic endothelial cells (LECs). The initial lymphatics have distinctive, discontinuous buttons in the endothelium, while the collecting vessels are covered with a continuous basement membrane and smooth muscle cells (SMCs). Endothelial cells in collecting vessels are an elongated shape, and they are connected by continuous zipper-like junctions [27]. Both types of junctions consist of proteins typical of adherens junctions (AJs) and tight junctions (TJs). The lymph flow is unidirectional by the function of intraluminal valves, which separate collecting lymphatic vessels to a series of functional units [27]. The high pressure of the lymph upstream of a valve enables lymph flow, which opens the valve, reverses flow-pushes of the leaflets against each other, and closes the valve as in the vasculature system [27]. Therefore, both the opening and closing of the valve depend on periodic changes in fluid pressure within collecting vessels, which are regulated by the smooth muscles via unmyelinated nerve fibers.

In the meninges of the brain, there are also two types of afferent lymphatic vessels: initial and collecting vessels. Anatomically and functionally, these vessels are different in the presence or absence of surrounding SMCs and lymphatic valves. The meningeal lymphatic vessels along the superior sagittal sinus are devoid of SMCs, lymphatic valves, and capillaries [4]. In the basal part of the skull of the brain, the lymphatic valves and capillaries of basal meningeal lymphatic vessels (mLVs) are revealed to locate adjacent to the subarachnoid space in mice. The basal mLVs facilitate the uptake and drainage of CSF [5,16]. Collecting lymphatic vessels are mostly associated with zipper-like LEC junctions, α-smooth muscle actin positive SMCs coverage, and lymphatic valves. In contrast, the basal mLVs possess a capillary network with loose button-like LEC junctions without SMC coverage, which optimizes the uptake of fluid and macromolecules [16]. According to Chen et al. (2020), after subarachnoid hemorrhage, meningeal lymphatics promote the clearing of the extravasated erythrocytes in CSF by draining them from CSF into the cervical lymph nodes, while the pathology of subarachnoid hemorrhage becomes further aggravated by the inhibition of vascular endothelial growth factor receptor 3, VEGFR3 [28]. VEGFC (Vascular endothelial growth factor-C) and VEGFD (Vascular endothelial growth factor-D) are the ligands of VEGFR3 tyrosine kinase. The meningeal lymphatics drain extravasated erythrocytes from CSF into cervical lymph nodes, and the inhibition of VEGFR3 exacerbates the pathology of subarachnoid hemorrhage [28].

Briefly, researchers use the marker proteins of the brain lymphatic vessels: LYVE-1, prospero homeobox protein 1 (Prox-1), PDPN (Podoplanin), VEGFR3, and chemokine (C-C motif) ligand 21 (CCL21). It is distinguishable from the expression of a blood vessel marker protein: CD31 (PECAM-1, platelet endothelial cell adhesion molecule-1) [4,5,21].

Importantly, the basal mLVs are hot spots for the CSF macromolecule clearance, which are located close to midbrain reticular nuclei, periaqueductal gray, anterior and posterior lobules of the cerebellum. The effects of dysfunction of lymphatic vasculature systems would be potentially severe in such regions. For instance, the ventral tegmental area includes lots of dopaminergic neurons and contributes to motivational behavior, and thereby, major depression disorder (MDD) and bipolar disorder (BD). The midbrain close to the periaqueductal gray is implicated in the expression of autonomic nervous system activity via serotonergic neurons, which regulate the sleep/wake rhythm as well as mode and cognition. The cerebellum is one of the hottest regions newly known related to distinct psychiatric disorders: autism spectrum disorders (ASDs), schizophrenia, and mood disorders [10,29,30,31], although the cerebellum had been considered the central area of motor coordination, and cerebellar abnormalities were believed to exclusively induce coordination disorder and tremor [32,33,34]. Therefore, the view of aberrant immune-triggered psychiatric disorders should be noticed what brain regions are functionally disrupted and what connections of brain regions are impaired. These combinations may produce various phenotypes of behaviors, and perhaps, the spectrum of psychiatric symptoms.

#### 2.1.3. Lymphatic Vasculogenesis

In the development of primitive blood vessels (i.e., endothelial tube), the circulatory system precedes the nervous system. In the early stage of development, hemangioblasts and angioblasts appear in the mesoderm. Hemangioblasts have the potential to differentiate into vascular endothelial cells and blood cells, while angioblasts are destined to differentiate only into vascular endothelial cells. The phenomenon in which blood vessels (endothelial tubes) are formed by the differentiation of hemangioblasts into vascular endothelial cells is called “vasculogenesis”. Endothelial cells become escorted by medial SMCs and pericytes, and perform morphological and functional differentiation into arteries, capillaries, and veins [35,36]. Lymphatic vasculogenesis is known to be mediated by transdifferentiation of venous endothelial cells toward the lymphatic endothelial phenotype [19,27,37]. In the developmental stages, the mammalian lymphatic vasculature is thought to exclusively stem from preexisting embryonic veins. An elegant lineage tracing study of the lymphangiogenesis by Srinivasan et al. (2007) demonstrated the venous origin of the mammalian lymphatic vasculature [38]. Another study suggested a significant contribution of non-venous-derived cells to the dermal lymphatic vasculature [39], although the studies of lymphatic vasculogenesis in the brain are just getting started.

In mice, LECs in the body are first specified in the anterior cardinal vein around embryonic day 9.5 (E9.5) when a subset of venous endothelial cells expresses the homeobox transcription factor Prox1 and the LYVE-1 [40]. The transcription factor SOX18 induces Prox1 expression and initiates LEC commitment in mice [40]. In vitro studies using human primary LEC and blood vascular endothelial cells (BECs) demonstrated SOX18 binding to the Prox1 promoter and showed that PROX1 confer BECs to lymphatic identity and direct them to LECs [40,41]. Thus, SOX18 and Prox1 constitute essential signaling for LEC differentiation. The Prox1-interacting nuclear receptor Coup-TFII has been shown as an earlier venous identity factor and regulates the expression of LEC-specific genes, such as neuropilin-2 [42]. In the early developmental stages, Prox1/LYVE-1–positive cells bud and migrate in the central veins. Subsequently, they would form the first bona fide lymphatic structures in regions where lymphangiogenic growth factor VEGFC expresses from the lateral mesoderm [27].

On the other hand, monocytes/macrophages have been proved to have involvement in lymphatic neoplasia in at least two ways: a source of VEGFC after appropriate stimulation or by transdifferentiation into LECs that integrate into the growing capillaries [19]. According to the studies of Kerjaschki and colleagues, de novo lymphangiogenesis was originally found in tumor-associated macrophages [43] and in kidney transplants during organ rejection [44]. The transdifferentiation of bone marrow-derived CD11b+ macrophages to LECs was also found by another laboratory with observation under inflamed conditions in the corneal stromata. Further, the systemic elimination of macrophages can prevent lymphangiogenesis [45]. Subsequent studies indicate that the adaptive immune system can regulate lymphangiogenesis in lymph nodes in both positive and negative directions, respectively [46,47]. More recently, in the CNS, Hsu et al. (2019) revealed that lymphatic vessels nearby the cribriform plate induce lymphangiogenesis in a VEGFC-VEGFR3 dependent manner during experimental autoimmune encephalomyelitis (EAE) as an animal model of multiple sclerosis (MS) [48]. The lymphatic vessels of the cribriform plate drain both CSF and immune cells in the CNS parenchyma. Neuroinflammation-induced lymphangiogenesis also promotes the drainage of CNS derived antigens that will proliferate antigen-specific T cells in the draining lymph nodes during EAE. In contrast, meningeal lymphatics did not undergo lymphangiogenesis during EAE, and authors claimed the distinct heterogeneity in CNS lymphatic vessel formations and consequent CNS immune surveillance [48].

### 2.2. Glia-Lymphatic System (Glymphatic System)

#### 2.2.1. Glymphatic System

In recent years, the glia-lymphatic system, also referred to as the glymphatic system, has been drawing more interest across various fields of medico-pharmaceutical neuroscience. This is owing to the recent discovery of the meningeal lymphatic system being closely implicated in a wide spectrum of neurological pathologies, such as Alexander’s disease (AxD), AD, Parkinson’s disease (PD), Huntington’s disease, amyotrophic lateral sclerosis (ALS), epilepsy, and various psychiatric conditions such as ASDs, and schizophrenia [11,49,50,51]. This insight has opened a new arena of tackling these conditions in search of potential novel cures.

The term glymphatic system was initially invented by a team led by Maiken Nedergaard, owing its name to the mechanism resembling the lymphatic system, in conjunction with glial cells [52] (Figure 2). It involves the convective flow of CSF mediated by abundant water channels called aquaporins; among subtypes, aquaporin-4 (AQP-4) is the most prevalent and relevant type in the CNS for water homeostasis [53,54,55]. AQP-4 channels are located most prominently on the vascular end-foot processes of astrocytes [2,56,57]. Astrocytes have been extensively studied in the past few decades, whereby some critical roles in the CNS and associated pathologies have been revealed. Astrocytes encompass many essential life-sustaining functions in the brain [56,57,58,59]. Some of those functions pertinent to the current topic include homeostasis of ion and glucose concentrations in the extracellular space, neuroendocrine regulation of the blood flow via vasomodulation, circadian rhythm, nutrients, as well as various immune cells and neurotransmitters in the CNS [52,56,57]. Astrocytes indirectly participate in the conduction of neuronal electrical activities, the plasticity of presynaptic release and postsynaptic responsiveness, and the regulation of electric circuits of neurons by releasing molecules. Interestingly, among them, LIF (leukemia inhibitory factor) is known to promote the myelination of oligodendrocytes [60]. Furthermore, astrocytes are activated in response to insults to the CNS in forms such as trauma and inflammatory processes accompanied by social stressors in the neurodegenerative diseases [50,61]. Via these pathways, astrocytes are capable of transforming themselves into a reactive state to facilitate repair and regeneration of the damaged CNS by altering the expression of some of the genes such as the glial fibrillary acidic protein (GFAP) genes. The resulting “astrogliosis,” an abnormal increase in the number of astrocytes, amplifies the release of astrocytic glutamate, which eventually influences the outcome of neuronal integrity and cytotoxicity [50,51,56,61,62,63,64]. Astrocytes are also unique in a way that they have the capacity to respond to nearly all neurotransmitters when activated [65].

For decades, it has been our common understanding that the metabolic activities in the brain well exceed that of any other organs’. Yet the fate of its metabolic byproducts such as amyloid-β, α-synuclein, and tau protein, which are described as the culprit of many devastating neurodegenerative diseases, has been points of controversies to date [55,66,67]. According to recent studies by Nedergaard et al. (2013 and 2020) [68,69], much of the housekeeping for such metabolic wastes from the interstitial space of the CNS parenchyma is carried out while our brain is in the resting state, more specifically, when the brain is in the non-REM sleep mode or is producing slow δ-waves in inverse proportion to heart rate [70,71,72].

#### 2.2.2. Astrocytic Contribution and AQP-4 Function

The bulk of the glymphatic systems is attributed to the functions of astrocytes; the aforementioned astrocytic water channel AQP-4 is the crucial mediator of the exchange of ISF and CSF, which overall acts much in the manner of a pseudolymphatic system, governed by circadian rhythm in the CNS [54,73,74]. Interestingly, AQP-4 channels not only play a critical role in the homeostasis in the CNS, but they also have a clinically significant supportive role in other sensory and metabolic organs [49,54,73,74]. The mean rate of CSF production is 0.4 mL per minute in healthy humans and 0.2 mL per minute in patients with AD [75]. In rats, the CSF production rate is 1.2 to 1.5 μL per minute in adults and halves in the aged group [76]. Interestingly, a recent study reported variability of the brain BOLD (blood oxygen level-dependent) signal in patients with AD, mainly due to the cardiovascular pulsations in the brain parenchyma [77]. The authors detected neither differences in the average cardiorespiratory rates nor the blood pressure between the control and AD groups [77]. Thus, the impairment of the brain vasculature system could be used as the biomarker for detecting AD patients.

In a review article, Mader and Brimberg (2019) laid out that AQP-4 may be a factor in a number of pathologies [73]. Some of the neurodegenerative diseases such as AxD, AD, ALS, PD, as well as stroke, trauma, and psychiatric diseases (e.g., schizophrenia, major depressive disorder, and ASD) are accompanied by dysfunctional AQP-4 and, thereby, the glymphatic system. It was claimed that there is a controversy regarding the clinical relevance of the astroglial AQP-4 functions in the role of water homeostasis; as it turned out, AQP-4 was not found to affect the barrier functions for macromolecules, nor the water content of the brain [11,78]. The following study by Iliff et al. (2012) [2] demonstrated that the fluorescence of small, but not large, molecular weight interstitial tracers were less spread in the AQP-4-null mouse brains than controls, suggesting the AQP-4-dependent interstitial solute and fluid clearance in the brain. The perivascular astroglial sheath is completely continuous and covering the capillary surface without gaps, and therefore, the AQP-4 would play a rate-limiting role in astrocytic endfeet [79].

It is now becoming evident that previously accepted models of CSF drainage were rather rigid and over-simplified, guided by conventional studies that were designed upon a set of assumptions due to various technical limitations [55]. Particularly, studies utilizing tracer distribution analyses were susceptible to artifacts [55]. Furthermore, unsurprisingly the large periarterial spaces were not visible on fixed tissues without physiological pulsatile circulation. Consequently, the conclusions drawn based upon postmortem studies were hence skewed, reflecting only the non-dynamic, fractional picture of the system [55,80,81]. The periarterial spaces were elucidated only after high-resolution in vivo imaging such as dynamic contrast-enhanced MRI (DCE-MRI) and two-photon laser scanning microscopy (2PLSM) became readily available [2,3,16,55,67,72,80,82,83,84]. DCE-MRI is an MRI perfusion technique that provides information pertaining to temporal perfusion of the system of interest with the administration of intrathecal paramagnetic contrast agents such as gadolinium [81,85]. 2PLSM is a fluorescence imagining technique based upon capturing the scattering and absorption patterns of the degradation signals by long-wavelength photons, which enables real-time dynamic analyses of a thick living tissue such as brain parenchyma including neurons, glia, and vasculatures [2,80,86,87,88,89]. A recent case report study showed the artificial opening of BBB using transcranial focused ultrasound in the human brain, which proved the system in humans and the persistence of glymphatic efflux in AD and ALS patients [90].

#### 2.2.3. Potential Pathophysiology of the Glymphatic System

As discussed earlier, the pathophysiology of those progressive neurodegenerative conditions such as AxD is closely attributed to dysfunctional astrocytes and impaired glymphatic system. In AxD, when the astrocytes are triggered likely by an inflammatory event, they cause the overexpression of the GFAP due to gain-of-function mutation [91,92,93]. This accumulation of falsely folded GFAP eventually aggregates to become Rosenthal fibers, which cannot be cleared by the innate glymphatic system [94,95]. This series of events, in turn, further upregulates gene expression, activating and recruiting more immune cells, and facilitating a cascade of a positive feedback loop [91,92,93]. It is therefore becoming increasingly evident that there is well-supported clinical relevance for exploring links among astrocytes, immune cells, microglia, and the glymphatic system.

Other evidence for the roles of the astrocytes and AQP-4 behind AD and epilepsy may also give more insights [96,97,98]. In the epileptic brain, the extracellular K^+^-buffering, AQP-4 localization, and domain organization of astrocyte processes are disrupted in mouse models and human patients [96,97]. Loss of perivascular AQP-4 might be involved in the pathogenesis of mesial temporal lobe epilepsy (MTLE). The density of AQP-4 along the perivascular membrane of astrocytes endfeet was reduced by 44% in hippocampal CA1 of MTLE compared to non-MTLE in human samples, suggesting leading to an impaired water and K^+^-homeostasis in MTLE parenchyma [96]. In mouse models of epilepsy, cortical astrocytes showed morphological changes and an increase in the overlap of their processes [97]. Therefore, mechanisms driving the CSF homeostasis are thought to be altered in epilepsy. Indeed, in contrast to AD patients, respiratory-related brain pulsations are increased in epilepsy patients without changes in physiological cardiorespiratory rates [99]. The most affected region of the brain was the upper brain stem respiratory pneumotaxic center, midbrain, and temporal lobes, including amygdala, hippocampus, globus pallidus, and putamen; almost of all of which are close to the cavernous sinus in epilepsy patients [99] (Figure 1).

### 2.3. Infiltration of Immune Cells to Brain across Vasculature System

#### 2.3.1. Immune Privilege

The existence of immune privilege had been recognized in the brain parenchyma and retina, even in the late 19th century [100]. The explanation of this phenomenon by Peter B Medawar (Medawar, 1948) [101] was that physical barriers around the immune-privileged site enabled it to avoid detection from the immune system: although the allografts are rapidly repelled from tissues such as the skin, they are accepted when placed in certain other sites, particularly the brain and anterior chamber of the eye. For instance, when excessive inflammation occurs in the eye in response to infection or external stimulants, the eye is equipped with an automatic control mechanism that suppresses inflammation to avoid visual dysfunction, so-called immune privilege. Recent results have revealed that different mechanisms of interaction with the immune system inhere not only in the eye and brain but also in the pregnant uterus and reproductive organs, which were previously thought of as immune-privileged sites [102]. The CNS was once thought to be immune-privileged sites that are not the target of the systemic inflammatory response or immune response. This was based on the observation that antigens and lymphocytes can barely cross the BBB for host (brain) defense and hardly elicit a cellular immune response to injuries and inflammation. However, the brain has been recognized to be able to initiate immune-mediated inflammatory responses, which contribute to the development of many neurodegenerative and cognitive diseases and, plausibly, other mental disorders.

#### 2.3.2. Glial Cell-Characteristics of the Brain and the Regulation by Immunity

According to von Bartheld et al. (2016), the validated isotropic fractionator demonstrates that glia:neuron ratios in the human brain is less than 1:1, although it was previously considered to be 10:1 [103]. The overall total number of neocortical neurons and glial cells was 49 and 65 billion in females and males, respectively, with a biological variance of 24% [104]. The investigation by Pelvig et al. (2008) suggested that the number of oligodendrocytes decreases by 27% throughout life from 18–93 years, which supports the idea that there is an interaction between neuronal activity and the number of oligodendrocytes. In fact, their results also suggest correlations between total numbers of neurons and oligodendrocytes, while the glia/neuron ratio over age was substantially constant [104]. The CNS consists of various types of cells: neurons and glia, and glial cells are separated into astrocytes, oligodendrocytes, and microglia. Below is a brief summary of those cell characteristics:

Astrocytes, “star-shaped cells” named by Mihály Lenhossék, are multifunctional. By asymmetric cell division, astrocytes arise from radial glial cells, the precursor cells of all types of neurons, astrocytes, and oligodendrocytes. Astrocytes substantially maintain the functional structure of the brain and regulate extracellular ion environment, BBB maintenance, neurotransmitter uptake, energy supply, growth of dendrites and extension of axons, repair of neural circuits, gliotransmitter release, Ca^2+^ oscillation, induction of synaptic and non-synaptic plasticity. They show various morphological forms: star-shaped, protoplasmic-types in the grey matter, and fibrillary type in the white matter. Tufted astrocytes and Rosenthal fibers are found in a pathological brain. Astrocytes were shown to integrate signals from various cytokines and chemokines (including interferon-γ, interleukin (IL)-17, IL-1β, IL-6, IL-10, and CCL2. GFAP, vimentin, AQP-4, excitatory amino acid transporter 2 (EAAT2), and S100β are the marker proteins that are expressed in those cells. They are also involved in metabolic processes, redox homeostasis, and brain development [105]. Reactive phenotypes (termed A1/A2 astrocytes, defined by upregulation of GFAP expression) may be toxic to neurons, releasing signals that can lead to neuronal death [62,63]. Bergmann glia in the cerebellum and Müller cells in the retina are astrocyte analogs.

Oligodendrocytes in the CNS produce the myelin sheath insulating neuronal axons, which allows for faster action potential propagation (increasing conduction velocity by at least 50 times [106,107]), which is intervened by the nodes of Ranvier. Local release of synaptic vesicles from axons promotes myelination via myelin basic protein [108]. A previous study has suggested that early social isolation results in the behavioral anomaly and cognitive dysfunction of mature animals that was correlated with white matter alterations, demyelination, and medial prefrontal cortex (mPFC) dysfunction [109], although the effect might be not restricted to the mPFC. Interestingly, in mouse mPFC oligodendrocytes, the effects of social isolation, beginning at postnatal day 21, was prominent to facilitate the demyelination. Such mPFC-dependent anomaly in social and cognitive behaviors occur only during a critical period between postnatal weeks three and five, which were not reversed by reintroduction to a social environment [109]. The maturation of progenitor cells to oligodendrocytes are also presumed to be suppressed by an immune challenge by activated microglia/monocytes in the early phenotype in schizophrenia [11]. Olig1 (Oligodendrocyte transcription factor 1), 2, and 3, myelin basic protein (MBP) is specifically expressed in the oligodendrocytes and used as marker proteins, whereas myelin proteolipid protein (PLP), sulfatide, and galactosylceramide are known as myelin proteins [110].

Microglia are the most abundant resident immune cells in the CNS and are characterized by the expression of IBA1 (ionized calcium-binding adaptor molecule 1) and CX3C-chemokine receptor 1 (known as fractalkine receptor). They promote synapse formation mediated by brain-derived neurotrophic factor BDNF [111]. Their contribution to neurodegenerative diseases, as well as their association with microbiota, is currently evident [112]. In contrast, their roles in chronic neuroinflammation are not entirely clarified yet [9]. After derived from primitive macrophages in the yolk sac, microglia colonize the CNS before embryonic day 9 (E9) [113], and they are alive for a long time there. In the periphery, macrophage populations are thought to be replenished by circulating monocytes derived from multipotent hematopoietic stem cells (HSCs) of the bone marrow, although this view has been challenged [114]. In contrast, experiments have shown that there is little infiltration of peripheral HSCs/monocytes/macrophages into the CNS to help maintain or replenish microglia under normal conditions due to the mature BBB [115,116,117]. In the disease model, monocytic infiltration is observed with progression to paralytic stages of EAE [115]. Parenchymal microglia modulate the intrinsic excitability [10,29,118] and the efficacy of synaptic transmission [119] of CNS neurons. They also engulf dendritic spines via C1q complement signaling in the developmental and senile brain [120,121]. A recent study using the Confetti (‘Microfetti’) mouse model for fate mapping indicated that microglia renew themselves randomly and steadily throughout the healthy CNS with regional differences in the rate of eight to 41 months [122]. The onset of neurodegeneration by injury triggers rapid clonal expansion and activation of microglial cells, while the BBB remains intact in their facial nerve axotomy [122]. The altered microglial network gradually returns to a steady-state cell density without macrophage emigration, which restores the CNS to a non-activated state [122]. Further, the transcriptomic analysis gives us important notions about microglia in the developmental program of gene expression patterns and chromatin dynamics [123], regional heterogeneity of gene profile [124,125], different renewal rate [122], and disease conditions, e.g., AD, glioma, and MS [126,127,128,129].

Non-microglia resident myeloid cells, bone-marrow-cell lineage, including perivascular, choroid plexus, and meningeal macrophages, and DCs also exist in the healthy and mature CNS parenchyma. Non-parenchymal meningeal and perivascular macrophages are of embryonic origin, i.e., ontogenetically related to microglia. In contrast, the choroid plexus macrophages are bone marrow-derived [9]. Myeloid cells are composed of different cell subsets, including DCs, macrophages, and monocytes. Microglia are the tissue-resident macrophages of the CNS parenchyma, which is distinct from most other macrophages. DCs and macrophages present antigen to infiltrating T cells in neuroinflammatory diseases. In the conditions of brain inflammation, tissue-invading monocytes are inflammatory DCs or monocyte-derived DCs/macrophages (Figure 3). Once inflammation commences, these cells are the major cell-type that presents antigen to infiltrating T cells in diseases like EAE. They disappear once the inflammation resolves [9].

#### 2.3.3. Infiltration of Immune Cells to Brain Parenchyma

In healthy brains, immune cells in the brain are known limited to infiltration into parenchyma because of the existence of the BBB. In principle, immune cells, including DCs and T cells, actively migrate from tissues to draining lymph nodes and are then returned to the blood circulation via the lymphatic vasculature for the purpose of immune surveillance [14]. LECs release extracellular chemokines to guide the migration of immune cells. Because of the lack of sufficient studies in the CNS, here, we mainly discuss the peripheral tissues. The tissue LECs are known to release CCL21, the signaling of which enables DCs and T cells to enter the lymphatic vasculature via chemokine receptor type 7 (CCR7) [135,136,137,138]. Under inflammatory conditions, tissue LECs also produce chemokine (C-X3-C motif) ligand 1 (CX3CL1) and chemokine (C-X-C motif) ligand 12 (CXCL12). CX3CL1 and CXCL12 facilitate the migration of DCs on the lymphatic vasculature via their receptors CX3CR1 and CXCR4, respectively [14,139]. In the CNS, CX3CR1 is the fractalkine receptor and is expressed on the surface of both microglia and macrophages in CNS. CX3CL1/CX3CR1 axis leads activated microglia in damaging oligodendrocytes, and thus the impairment of oligodendrocytes and white matter injury [140]. A recent report indicates that tissue LECs release exosome-rich endothelial vesicles, including a number of inflammatory cytokines in human chronic inflammatory diseases [141]. The secretion of exosome-rich endothelial vesicles results in the exploratory migration of immune cells and promotes directional migration of CX3CR1-expressing cells [141]. Intercellular adhesion molecule 1 (ICAM-1), vascular cell adhesion molecule 1 (VCAM-1), and E-selectin are also expressed in the LECs. They are involved in the adhesion and entry of the immune cells in dermal tissues and thus, accelerate the immune responses [14,142,143]. Johnson et al. (2017) unveiled that the endothelial hyaluronan receptor LYVE-1 is essential for DC-trafficking. While DCs are docked to the basolateral surface of lymphatic vessels, the migration of DCs to the lumen occurs through hyaluronan-mediated interactions with LYVE-1 [144]. Further experiments regarding molecular mechanisms of immune-cell infiltration in the brain should expand our knowledge.

Of note, in the lymphatic node, subcapsular sinus macrophages are yolk-sac-derived in the embryo and are maintained by colony-stimulating factor-1 produced by LECs in lymph nodes. When pathogen induces transient deletion of bone marrow-derived macrophages, they restore the subcapsular sinus macrophage network [145]. In contrast, in the CNS, microglia renew themselves under steady-state conditions, while they replenish through the region-specific clonal expansion after a pathological condition [117,122].

Severe inflammation in the brain induces damage to blood vessels, glymphatic system, and will cause neurodegeneration [9] (Figure 3). Severe inflammation of the brain can result in angiopathy, i.e., the necrotic activity of tissues, capillaries, and lymph vessels. Damages of blood vessels in the brain are the most popular in strokes (e.g., cerebral and cerebellar infarction, micro hemorrhage that ruptures blood vessels or capillaries in the brain, and subarachnoid hemorrhage), but at a microscopic level, damaged vessels/capillaries and the disruption of lymphatic vessels in the brain should be related to neurological and neurodegenerative diseases. BBB breakdown is one of the most critical concepts to interpret the emergence of cognitive and neurodegenerative diseases in the senile brain. As inferred from the mouse EAE model, the CNS inflammation and disruption of the barrier structure can occur in BBB breakdown in humans [15]. Damage to the BBB in AD is observed by the post-mortem human brain, while it is important to know when the BBB breakdown occurs. Current advanced MRI scanning and the direct-current electroencephalography enable the clinical diagnostic test to detect such BBB breakdowns [83,84,146].

#### 2.3.4. The Structure and Dysfunction of the BBB

Endothelial cells of capillaries are physically connected with each other through the TJ, AJ, and gap junction proteins (Figure 2). Endothelial TJ and AJ proteins themselves constitute the physical barrier of the BBB, preventing entry of leukocyte adhesion molecules, non-selective fenestrae, pinocytosis, and bulk-flow transcytosis [12,13] (Figure 3). The endothelium allows the rapid free diffusion of oxygen from the blood to the brain parenchyma and carbon dioxide to blood, which retains brain metabolism and adequate pH levels in the ISF and parenchymal cells. Only small lipophilic molecules (e.g., hormones) and drugs with a molecular weight of <400 Da and form less than eight total hydrogen bonds can cross the BBB [147,148,149]. It has been noted that 98% of small-molecule drugs fail in clinical trials due to less BBB permeability. Only limited opiates, anxiolytics (e.g., benzodiazepines), antipsychotics (e.g., chlorpromazine), and anti-dementia drugs (e.g., donepezil, memantine, and tacrine) can cross the BBB; thus, the efficacy of such drug treatment is limited [148,150]. It is worth mentioning that such a poor parenchymal uptake of agents from the CSF via BBB and glymphatic fluid transport are now overcome by the transcranial ultrasound [151,152]. Ultrasound protocol significantly improves the brain parenchymal uptake of drugs and antibodies (i.e., by around four times in case of 1 kDa MRI tracer signals) via intrathecal administration [151].

Pericytes are derived from the neural crest and are integrated into the endothelial and astrocyte functions at the neurovascular unit, and they regulate the BBB [153]. Pericytes appear to form a continuum from arteriole to venule and share a basement membrane with endothelial cells. Direct contacts of pericytes with endothelium are via N-cadherin and connexins [154]. They are also implicated in the angiogenesis and microvascular stability [154] and phagocytosing toxic metabolites [155]. According to Zhao et al. (2015) [13], pericyte degeneration and injury occur in many neurological diseases, including AD [156,157], mild dementia [83], ALS [158], and stroke [159].

The process of signaling between astrocytes and pericytes has been shown to deteriorate the integrity of the BBB. Studies of transgenic apolipoprotein E (APOE) mice have shown that APOE4, a major genetic risk factor for AD, leads to disruption of BBB integrity by activating a pathway: proinflammatory cyclophilin-A (CypA)—nuclear factor κB (NFκB)—matrix metalloproteinase 9 (MMP-9) in pericytes [134]. This pathway leads to degradation of the basement membrane and TJ proteins, resulting in chronic BBB breakdown, neuronal dysfunction, and secondary neurodegenerative changes [160]. In contrast, APOE3 but not APOE4 binds to the low-density lipoprotein receptor-related protein 1 (LRP1) in pericytes, which suppresses the CypA—NFκB—MMP-9 pathway [160].

Chronic systemic inflammation is the result of the chronic activation of the innate immune system, mediated by their release of pro-inflammatory cytokines (Figure 4). Experimental studies have indicated that systemic inflammation in preterm infants can alter the maturation of the developing brains and the behaviors of grown animals. The major proinflammatory cytokines responsible for early responses are IL-1β, IL-6, and tumor necrosis factor-alpha (TNF-α). Microglial TNF-α induces long-term potentiation (LTP) of intrinsic excitability of cerebellar Purkinje neurons [118], although the effect is brain region- and cell type-dependent. TNF-α also modulates the synaptic efficacy [118,161] and gliotransmitter release [162]. IL-1β prevents the LTP induction of population spikes in the hippocampal CA1 [163], although inhibition of IL-1β prevents its maintenance [164]. Mice exposed to IL-1β had a long-lasting impairment of myelination of oligodendrocytes, which resulted in an increase in nonmyelinated axons while accompanied by an increased density of immature oligodendrocytes. These morphological abnormalities were accompanied by a reduction of white-matter substrate and memory deficits [165]. In a previous culture study, lipopolysaccharide-induced microglial activation impeded oligodendrocyte lineage progression, reduced the production of MBP and resulted in the death of oligodendrocyte progenitor cells (OPCs). Those processes involved nitric oxide-dependent oxidative damage and the release of TNF-α and pro-nerve growth factor [166]. Microglial activation by methotrexate (a chemotherapy reagent to treat cancer, autoimmune diseases, rheumatoid, and ectopic pregnancy) leads to persistent disruption of oligodendrocyte lineage dynamics and astrocyte reactivity, resulting in the depletion of white matter OPCs, the persistent deficit in myelination, and the chemotherapy-related cognitive impairment [167] (Figure 4). In humans who received chemotherapy, persistent depletion of oligodendrocyte lineage cells has been observed [167]. Recent in vivo mouse studies showed that the rapid perivascular microglial clustering triggered by fibrinogen leakage upon BBB disruption initiates the development of axonal damage in the EAE model as neuroinflammation [89]. Also, systemic inflammation induces CCR5-dependent migration of brain resident microglia to the cerebral vasculature and microglia to phagocytose astrocytic endfeet during sustained inflammation, resulting in the impairment of BBB function [168].

As aforementioned, the BBB is a selective capillary barrier for solutes to enter the CNS. The leukocyte recruitment to the CNS and the spatial separation is conducted through two steps: first, leukocytes can pass across post-capillary venules into the perivascular space (i.e., Virchow–Robin space) (Figure 2). Under inflammatory conditions, a wide range of leukocytes can enter this space, but this is not necessarily related to pathology, as Becher et al. has pointed out [9]. Second, leukocytes progress across the glia limitans, a barrier formed by astrocytic endfeet into the neuropile, which locates over the brain parenchyma [9] (Figure 3 and Figure 4). Of note, the link between the immune-cellular infiltration and BBB dysfunction is under debate. Previous studies in mice with EAE suggested that the areas of enhanced BBB permeability does not correlate with the site of parenchymal infiltration [173]. In fact, in a number of neurological diseases, including AD, prion diseases, and Acquired Immunodeficiency Syndrome (AIDS)-related dementia, it had been postulated that leukocytes trespass on the brain parenchyma without the BBB breaking down [173]. In the adult rat, injection of IL-1β induced a pronounced increase in the permeability of the BCSFB solely in the meninges, but not in vasculatures [174]. In contrast, the higher permeability of the BCSFB after inflammation in the younger brain (three-week-old rat) is distinct from that of the adult brain [174], which is assumed that neurological consequences via BBB leakage after brain inflammation is utterly different in the course of brain development and astrocyte maturation. The leukocyte recruitment in the brain would be mediated by IL-1β or de novo synthesis of additional IL-1β but not TNFα, depending on the regression of the tight junction proteins: occludin and ZO-1 (Zonula occludens-1), and the injection of N-Methyl-D-aspartate (NMDA), an agonist of excitatory NMDA-type glutamatergic receptors, and endotoxin lipopolysaccharide (LPS) exacerbates the inflammatory recruitment [175,176]. Thus, those previous studies provided suggestive insight. Very recently, SARS-coronavirus 2 (SARS-CoV-2) infection was revealed to invade host cells via angiotensin-converting enzyme 2 (ACE2) expressed in the choroid plexus epithelium, causing the leakage of the brain barrier [177,178]. More severe damages and dysfunctions are expected in the brain regions close to the leakage.

During steady-state immune-surveillance, leukocyte extravasation across the BBB is limited to few activated T lymphocytes that interact with ICAM-1 and VCAM-1 expressed on the lumen of vascular endothelial cells. CXCL12 expression by endothelial cells on the abluminal side promotes activated CD4+ T cells to penetrate the perivascular space. After extravasation, T cells can interact with perivascular macrophages (PVM), a heterogeneous group of macrophages in the perivascular space, perform crucial activities for the maintenance of tight junctions between endothelial cells and limit vessel permeability, and phagocytoses of pathogens to avoid inappropriate inflammation (Figure 3). In inflammatory diseases, PVMs respond to cytokines, such as angiopoietin 2, macrophage colony-stimulating factor-1, and CXCL12. PVMs detach from capillaries and decrease their release of pigment epithelium-derived factor (PEDF), which results in a decrease in stability of tight junctions between blood vessel endothelial cells and increased vascular permeability. PVMs receive circulating immune complexes through Fcγ receptor IV (FcγRIV), which induces inflammation, recruitment of monocytes, and neutrophils into the interstitium [130]. During the parenchymal inflammation in AD and MS, infiltration of T cells to the parenchyma and the white matter is also observed [131,132,133].

Many acute and chronic inflammatory disorders in the brain involve inflammation of both the parenchyma and the vasculatures. Lastly, we sum up present BBB leakages of current studies in the condition of stroke [179], cerebral ischemia [180], sepsis [181,182], Malaria [183,184], intracellular parasite [185], SARS-CoV-2 [177,178], MS [186], systemic lupus erythematosus (SLE) [187], and even AD [188] and EAE [89]. Other viral infections and parasite invasion would increase the risk [189,190,191]. Therefore, many brain inflammation-associated diseases may cause leukocyte infiltration and associated behavioral anomalies.

### 2.4. Implications of the Inflammation and Immunity for the Pathophysiology of Psychiatric Disorders

As discussed in Khandaker et al. (2015) [189], a possible link between schizophrenia, the immune system, and infections was postulated over a century ago. Epidemiological and genetic studies suggested links with infection and inflammation [189]. In recent decades, experimental and diagnostic studies have indicated complex interactions between the immune system, inflammation, brain functions, and behavior. Here, we hypothesize that the dysfunction of vasculature systems and aberrant immunity may cause severe behavioral characteristics and dysfunction of neurons, astrocytes, and oligodendrocytes (Figure 4). The dysfunction would induce the malfunctions, exemplified hyper-, and hypo-connection of brain functional connectivity [118]. Such an idea is not new and has already been provided by other studies [192,193,194]. However, it did not advance because the brain lymphatic systems were not well-known at the time. In this last section, we describe some of the vital evidence of research regarding aberrant immune response in various types of psychiatric disorders and possibly related vascular dysfunctions that researchers in this field and psychiatric clinicians might be interested in.

In the past clinical investigations of schizophrenic patients showed no differences from the normal young males for cerebral blood flow and oxygen consumption [195]. However, in the late 20th century, cerebral metabolic hypofrontality, a state of decreased cerebral blood flow (CBF), has been found in schizophrenia patients, as well as MDD [196]. It gives an assumption that both the utilization of glucose and blood flow in the prefrontal cortex is reduced in those psychiatric disorders. Following studies showed the CBF changes in the cingulate cortex, thalamus, basal ganglia, and cerebellum in schizophrenic patients [197,198,199]. Since the end of the 20th century, researchers have also noticed the increase and decrease in the number of immune cells and inflammatory cytokines. For instance, ASD, schizophrenia, and MDD are relevant to the anomaly in their immunity [200,201]. In ASD patients, the blood plasma cytokine levels of IL-1b, IL-2, IL-6, IL-8, and TNF-a are all increased [200,202,203]. In a mouse model of ASD, an increase in the inflammatory cytokines (: IL-6, IL-12p40, CXCL10, CCL2, 3, and 4) have been correlated with the frequency or extent of the autistic-like behavior [204]. Maternal immune activation (MIA) in mice also shows ASD-related behaviors, representing infection-mediated neurodevelopmental disorders. A recent study showed the behavioral anomaly of MIA offspring is dependent on the IL-17a pathway [205]. The significance of the increased IL-6 signaling in MDD has also been discussed elsewhere [201,206]. Patients with BD show high levels of soluble IL-6 receptors and TNF receptors in serum samples [207], and the levels of proinflammatory cytokines have been correlated to the reduction of gray matter volume [207]. Maternal inflammation during the third trimester of pregnancy increased the plasma concentration of IL-6, TNF-α, and CCL2, and the rate of the attention-deficit hyperactivity disorder of the birthed children [208]. Thus, many current studies suggest the risk of aberrant immunity on the emergence of psychosis symptoms. Furthermore, an analytical study by Taoka et al. (2017) [209] suggested the impairment of the glymphatic system in AD using diffusion tensor image analysis [209]. The breakdown and disruption of brain immune-vascular systems are now investigated in the AD and ALS [83,84,90,146], although studies on other types of psychosis in animal models and human patients are expected to follow. Variability in psychiatric disorders also comes from genes and other intrinsic factors. Those complex factors may decide the specific vulnerability in each brain region and the resultant behavioral anomaly [31]. We also discussed the infiltration of immune cells into the brain parenchyma and the physiological disruption of neurons and glia in an earlier section. This schema could be implicated in the infected or post-infected brain.

## 3. Conclusions

Aberrant immunity of the brain parenchyma appears to be implicated in the disruption of normal neurophysiology and brain function, which can lead to changes in mood, cognition, behavior, and thought throughout life; in the course of developmental stages, in aged brains, and in the disease conditions [11,31,210]. Inflammatory cytokines released from the immune cells are received via receptors of neurons in the CNS and spinal neurons [161,169,170,171,172], in which the various forms of the plasticity of the efficacy of synaptic transmission and the intrinsic excitability of their membrane, even in the dendrites, are induced [118,211,212,213,214].

The traditional view of an immune privilege shielded by the BBB is now challenged by the observations of the breakdown and an increase in the immune-cell permeability from the blood vessels and lymphatic system at a high immune-responsive state in the brain, which we discussed throughout this review. These processes: infiltration of immune cells into the parenchyma, the release of inflammatory cytokines, destruction of the glymphatic system formed by astrocytes, breakdown of the BBB, impairment of oligodendrocyte maturation and demyelination, and resultant damages or apoptotic events, are suggested to drive the dysfunction of the entire brain activity to cause abnormality in the functional connectivity, and to aggravate the psychosis symptoms. This hypothesis challenges the longstanding neuron-centric view of neurodegenerative and psychiatric diseases and agrees with the gliocentric view [11,167]. However, we are not yet able to answer the points below: When is immune privilege established and become sound? Where are the vulnerable areas of the brain prone to neurophysiological dysfunction? (This should be related to why various types of symptoms emerge in psychiatric diseases and why different types of diseases share the symptoms.) Is there any relevance to metabolism in the brain? What is the relevance to entire brain dysfunction as functional hyper- and hypo-connectivity? How much is transcriptome and proteome data useful for understanding such physiological destructions? How can we develop treatments to prevent or cure the progression of psychotic symptoms? Therefore, our efforts to tackle these questions should enable us to overcome the symptoms.

## Figures and Tables

**Figure 1 biology-10-00034-f001:**
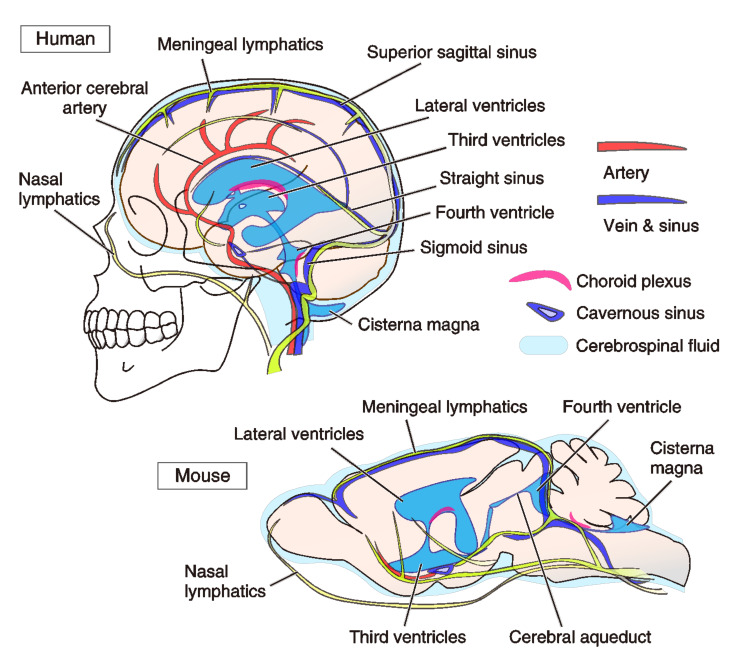
Anatomy of vasculature systems in human and mouse brains. Lymphatic and blood vessels in the human and mouse brains are shown. The cerebrospinal fluid (CSF) is produced in the choroid plexus of the ventricles. CSF circulates through thin lymphatic ducts in meninges. Pulsation mechanisms substantially drive the flow stream of CSF and meningeal lymph. We provide the whole lymphatic system in the brains, within our current knowledge, and their relationship to other organs, such as blood vessels and ventricles. According to Absinta et al. (2017) [8], in the human brain, the brain lymphatic vessels (LVs) are found in the superior sagittal sinus, straight sinus, cingulate gyrus, and middle meningeal artery. Meningeal LVs bilaterally run in parallel to the veins and sinuses. Additionally, LVs are found in cribriform plate at the initiation of the nasal lymphatics. The size of vessels is from 30 μm to 300 μm in diameter, depending on the location. In the marmoset brain, LVs are found in the superior sagittal sinus, and straight sinus [8]. According to Aspelund et al. (2015) [5], in the mouse brain, LVs are found in the rostral rhinal vein, superior sagittal sinus, transverse vein, sigmoid sinus, retroglenoid vein, middle meningeal artery, and pterygopalatine artery, in addition to the cribriform plate with a 10–40 μm diameter [5].

**Figure 2 biology-10-00034-f002:**
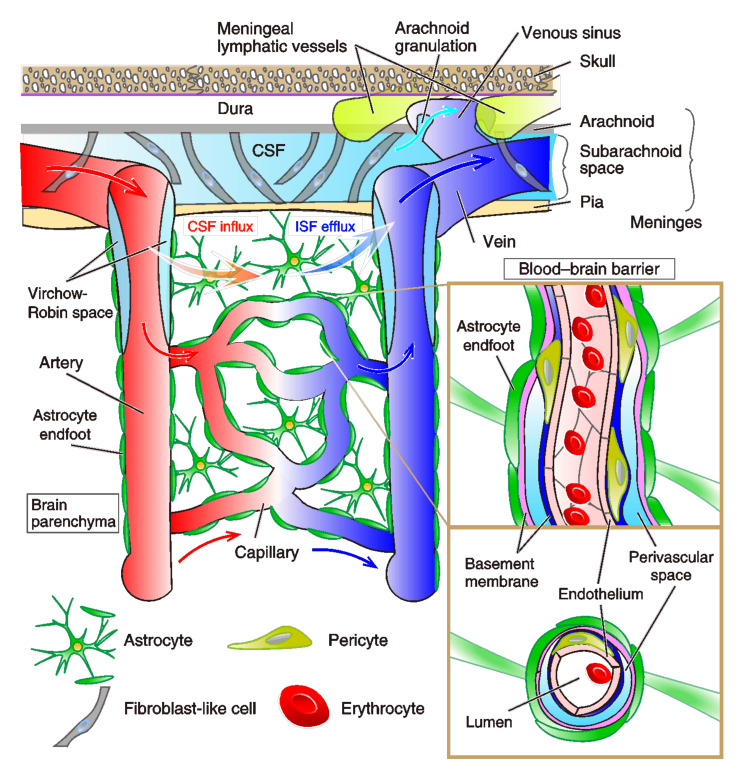
Schematic cytoarchitecture of cortical and meningeal vasculatures. Leptomeningeal arteries vertically penetrate the cortical parenchyma and branch into arterioles, capillaries, and postcapillary venules, which lastly converge into cortical veins. In the subarachnoid space, fibroblast-like cells and a collagen layer surround the arteries and veins. Cerebrospinal fluid (CSF) circulates right under the dura in the subarachnoid space. Smooth muscle cells cover the penetrating arteries, and pericytes are located around arterioles. Perivascular spaces that surround penetrating vessels are specifically called Virchow-Robin spaces, which are filled with pia-associated interstitial fluid (ISF). In capillaries, endothelial cells (ECs) compose the endothelium, which is coated by basement membranes (*inset*, enlarged images of vertical and transverse sections). Around the postcapillary venules, there is another fluid space, called a perivascular space, between the astrocytic basement membrane (rose-pink) adjunct to endfeet of astrocytes and the endothelial-cell basement membrane (navy blue). This organized structure is the blood-brain barrier (BBB). Blood vessels are made up of ECs and mural cells on the abluminal surface of the endothelial cell layer. Adjoined ECs are joined at tight and adhesion junctions. The BBB is formed by ECs, pericytes embedded in the capillary basement membrane, and astrocyte endfeet that enclose the capillary in a sheath. Postcapillary venules terminate to veins, which exit the CNS parenchyma. In the dura mater, both meningeal lymphatics (light green) and fenestrated blood vessels devoid of tight junctions lie down. Meningeal lymphatic vessels drain macromolecules from the CSF. Arachnoid granulations (Pacchionian granulations) are mulberry-like small protrusions of the arachnoid mater into the dura mater. They protrude into the dural venous sinuses and allow CSF to exit the subarachnoid space and to enter the bloodstream via veins (i.e., transverse sinus and sigmoid sinus), all of which terminate to the internal jugular vein. The glymphatic system is a pathway for the brain perfusion by CSF/ISF for the waste clearance (orange and blue gradient arrows). At the para-arterial vascular space between the basement membrane of smooth muscle cells and glia limitans, the CSF influxes into the brain parenchyma, which is coupled to a clearance mechanism for the removal of ISF and extracellular solutes from the interstitial compartments of the parenchyma. The water component of CSF crosses the astrocytic aquaporin-4 (AQP-4) water channels (expressed at the side of perivascular space of astrocyte endfeet), and the solutes enter the brain parenchyma via astrocytic transporters or ion-channels. Exchanged ISF passes through the perivenous space, driven primarily by arterial pulsation. The capillaries and basal meningeal lymphatic vessels are morphologically correlated; the capillaries have a peripheral morphology against the large blood vessels, while the basal meningeal lymphatic vessels also have a terminal/initial morphology against the large lymph vessels.

**Figure 3 biology-10-00034-f003:**
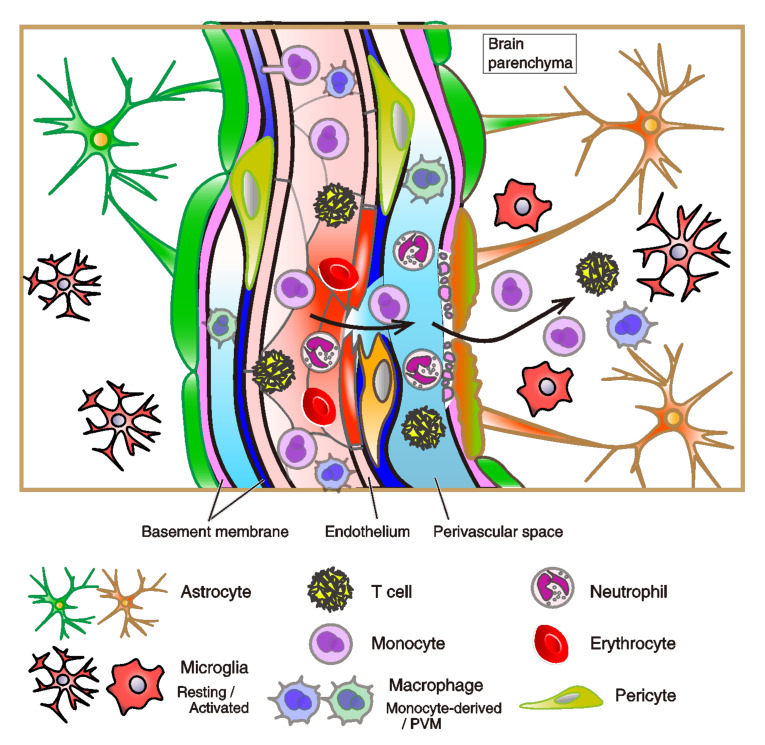
Breakdown of the central nervous system (CNS) vasculatures during inflammation. The schema of breakdown of the blood vasculatures and glia limitans. The disruption of the blood-brain barrier (BBB) and immune-cell permeation in the brain parenchyma are shown. Here, we suppose immune-pathophysiological state in the brain upon steady-state immune surveillance and inflammation, caused by vascular injury and cerebral hemorrhage, infection of viruses or microorganisms, neurological diseases, and other external factors. In the inflammatory condition, perivascular macrophages (PVMs) respond to cytokines, such as angiopoietin 2, macrophage colony-stimulating factor-1, and CXC chemokine ligand 12 (CXCL12). PVMs detach from capillaries and reduce their release of pigment epithelium-derived factor (PEDF), which destabilizes the tight junctions between blood vessel endothelial cells and increases vascular permeability. PVMs receive circulating immune complexes through Fcγ receptor IV (FcγRIV), which induces inflammation, recruitment of monocytes, and neutrophils into the interstitium [130]. During inflammation in the parenchyma, T cells infiltrate into parenchyma and white matter, and they release interferon-γ [131,132,133]. APOE4 is also implicated in the disruption of the BBB by stimulating a pathway: proinflammatory cyclophilin-A (CypA)—nuclear factor κB (NFκB)—matrix metalloproteinase 9 (MMP-9) in pericytes in AD brains [134]. Mural cells and other leucocytes (e.g., dendritic cells and B cells) are omitted from the figure. The subfamily of T cells: helper, killer, effector, and regulatory T cells is not subdivided.

**Figure 4 biology-10-00034-f004:**
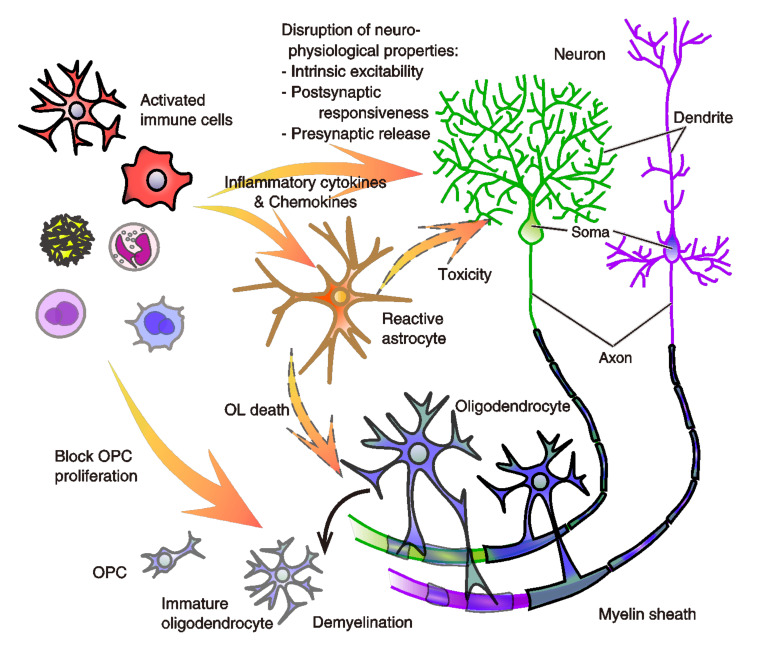
A model of aberrant immunity-driven neurophysiological disruption in the emergence of psychosis symptoms. Infiltration of stimulated immune cells into the parenchyma causes the disruption of various physiological properties of neurons there. They cause a hyper- and hypo-excitability of the intrinsic membrane properties decided by the function of K^+^, Na^+^, and Ca^2+^ channels. Among them, Ca^2+^-activated and voltage-sensitive K^+^ channels (e.g., small conductance Ca^2+^-activated K^+^ (SK) channel, and hyperpolarization-activated cyclic nucleotide-gated (HCN) channel) of neurons are shown to be modulated by inflammatory cytokines. Modulation of anion reversal potential is also indicated [169]. The other property of the postsynaptic responsiveness, decided by the expression or function of the postsynaptic ionotropic receptors, also changes in response to the release of inflammatory cytokines via cytokine receptors of neurons. It is also shown that the aberrant activity of immune cells also modulates presynaptic transmitters’ release of neurons [10,31,118,119,161,169,170,171,172]. Further, while oligodendrocytes make the myelin sheath of neuronal axons, which enables neurons to conduct action potentials efficiently to communicate with each other by insulating them, they are impaired by aberrant microglial activity. Demyelination and deficits in oligodendrocyte maturation cause structural hypomyelination and a loss of function of the white matter. Abnormality of astrocytic differentiation also occurs, and it reduces synaptic coverage of astrocytes. Not only the link of disrupted glial-cell differentiation to the initial event in schizophrenia [11], we also propose the relevance to pervasive psychosis symptoms observed in many psychiatric disorders [11,140,166,167]. These neurophysiological dysfunctions occurring at the parenchymal and white-matter areas by the immune-cell infiltration and aberrant immune-cell activation, especially at the vicinity of vascular and lymphatic pathways, may produce abnormal functional connectivity among brain areas. Abbreviations: OL, oligodendrocyte. OPC, oligodendrocyte precursor cell.

## Data Availability

Not applicable.

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
