# Peer review of "A Destruction Model of the Vascular and Lymphatic Systems in the Emergence of Psychiatric Symptoms"

_biology, 2021, doi:10.3390/biology10010034_

Round 1

Reviewer 1 Report

The manuscript is and extensive review about glymphatics and their potential role in immunity and inflammation.

The Authors must be commended for presenting a significant amount of previous data in a very detailed way.

However, before their contribution would be considered for publication, I would reccommend to include a section regarding the pathophysiological basis of beahviour disorders. There is a significant unbalance between the amount of presented data regarding glymphatics and pathophysiology of pesychiatric conditions. Therefore, the final assumtpion of psychiatric conditions possibly depending on a derangement of tissutal homeostasis is somewaht weakly supported by presented data.

I would also reccommend a general improvement of english Language throughout the manuscript.

Author Response

Thank you so much for your comment. I give responses mainly to your following suggestions:

> I would reccommend to include a section regarding the pathophysiological basis of beahviour disorders.

> There is a significant unbalance between the amount of presented data regarding glymphatics and pathophysiology of pesychiatric conditions.

We appended a new chapter indicating the link between the vascular, lymphatic systems, and the emergence of psychiatric disorders.

(New lines 571-)

  1. iv) Implications of the inflammation and immunity for the pathophysiology of psychiatric disorders

with additional citations:

Khandaker, G.M.; Cousins, L.; Deakin, J.; Lennox, B.R.; Yolken, R.; Jones, P.B. Inflammation and immunity in schizophrenia: implications for pathophysiology and treatment. Lancet Psychiatry 2015, 2, 258–270. doi: 10.1016/S2215-0366(14)00122-9.

Yolken, R.H.; Torrey, E.F. Viruses, schizophrenia, and bipolar disorder. Clin Microbiol Rev. 1995, 8, 131–145. doi: 10.1128/CMR.8.1.131-145.1995.

Müller, N.; Ackenheil, M. Psychoneuroimmunology and the cytokine action in the CNS: implications for psychiatric disorders. Prog Neuropsychopharmacol Biol Psychiatry. 1998, 22, 1–33. doi: 10.1016/s0278-5846(97)00179-6.

Hanson, D.R.; Gottesman, I.I. Theories of schizophrenia: a genetic-inflammatory-vascular synthesis. BMC Med Genet 2005, 6, 7. doi: 10.1186/1471-2350-6-7.

Kety, S.S.; Woodford, R.B.; Harmel, M.H.; Freyman, F.A.; Appel, K.E.; Schmidt, C.F. Cerebral blood blow and metabolism in schizophrenia. Am J Psychiatry. 1948, 104, 765–770.

Weinberger, D.R.; Berman, K.F. Speculation on the meaning of cerebral metabolic hypofrontality in schizophrenia. Schizophr Bull. 1988, 14, 157–168.

Ashton, L.; Barnes, A.; Livingston, M.; Wyper, D.; Scottish Schizophrenia Research Group. Cingulate abnormalities associated with PANSS negative scores in first episode schizophrenia. Behav Neurol. 2000, 12(1-2), 93–101. doi: 10.1155/2000/913731.

Franck, N.; O'Leary, D.S.; Flaum, M.; Hichwa, R.D.; Andreasen, N.C. Cerebral blood flow changes associated with Schneiderian first-rank symptoms in schizophrenia. J Neuropsychiatry Clin Neurosci. 2002, 14, 277–282. doi: 10.1176/jnp.14.3.277.

Paradiso, S.; Andreasen, N.C.; Crespo-Facorro, B.; O'Leary, D.S.; Watkins, G.L.; Boles Ponto, L.L.; Hichwa, R.D. Emotions in unmedicated patients with schizophrenia during evaluation with positron emission tomography. Am J Psychiatry. 2003, 160, 1775-1783. doi: 10.1176/appi.ajp.160.10.1775.

Prata, J.; Santos, S.G.; Almeida, M.I.; Coelho, R.; Barbosa M.A. Bridging Autism Spectrum Disorders and Schizophrenia through inflammation and biomarkers - pre-clinical and clinical investigations. J Neuroinflammation. 2017, 14, 179. doi: 10.1186/s12974-017-0938-y.

Ashwood, P.; Krakowiak, P.; Hertz-Picciotto, I.; Hansen, R.; Pessah, I.; Van de Water, J. Elevated plasma cytokines in autism spectrum disorders provide evidence of immune dysfunction and are associated with impaired behavioral outcome. Brain Behav Immun. 2011, 25, 40–45. doi: 1-.1016/j.bbi.2010.08.003.

Yamauchi, T.; Makinodan, M.; Toritsuka, M.;Okumura, K.; Kayashima, Y.; Ishida, R.; Kishimoto, N.; Takahashi, M.; Komori, T.; Yamaguchi,Y.; Takada, R.; Yamamuro, K.; Kimoto, S.; Yasuda, Y.; Hashimoto, R.; Kishimoto, T. TNF-α expression ration of M1/M2 macrophages is a potential adjunctive tool for the diagnosis of autism spectrum disorder. J Neuroinflamm. 2020, in press

Choi, G.B.; Yim, Y.S.; Wong, H.; Kim, S.; Kim, H.; Kim, S.V.; Hoeffer, C.A.; Littman, D.R.; Huh, J.R. The maternal interleukin-17a pathway in mice promotes autism-like phenotypes in offspring. Science 2016, 351, 933–939. doi: 10.1126/science.aad0314.

Onore, C.E.; Careaga, M.; Babineau, B.A.; Schwartzer, J.J.; Berman, R.F.; Ashwood, P. Inflammatory macrophage phenotype in BTBR T+tf/J mice. Front Neurosci. 2013, 7, 158. doi: 10.3389/fnins.2013.00158.

Maes, M.; Meltzer, H.Y.; Bosmans, E.; Bergmans, R.; Vandoolaeghe, E.; Ranjan, R.; Desnyder, R. Increased plasma concentrations of interleukin-6, soluble interleukin-6, soluble interleukin-2 and transferrin receptor in major depression. J Affect Disord. 1995, 34, 301–309. doi: 10.1016/0165-0327(95)00028-l.

Ting, E.Y-C.; Yang, A.C.; Tsai, S-J. Role of Interleukin-6 in Depressive Disorder. Int J Mol Sci. 2020, 21, 2194. doi: 10.3390/ijms21062194.

Bai, Y-M.; Chen, M-H.; Hsu, J-W.; Huang, K-L.; Tu, P-C.; Chang, W-C.; Su, T-P.; Li, C.T.; Lin, W-C.; Tsai, S-J. A comparison study of metabolic profiles, immunity, and brain gray matter volumes between patients with bipolar disorder and depressive disorder. J Neuroinflammation. 2020, 17, 42. doi: 10.1186/s12974-020-1724-9.

Gustafsson, H.C.; Sullivan, E.L.; Battison, E.A.J.; Holton, K.F.; Graham, A.M.; Karalunas, S.L.; Fair, D.A.; Loftis, J.M.; Nigg, J.T. Evaluation of maternal inflammation as a marker of future offspring ADHD symptoms: A prospective investigation. Brain Behav Immun. 2020, 89, 350–356. doi: 10.1016/j.bbi.2020.07.019.

Taoka, T.; Masutani, Y.; Kawai, H.; Nakane, T.; Matsuoka, K.; Yasuno, F.; Kishimoto, T.; Naganawa, S. Evaluation of glymphatic system activity with the diffusion MR technique: diffusion tensor image analysis along the perivascular space (DTI-ALPS) in Alzheimer's disease cases. Jpn J Radiol. 2017, 35, 172–178. doi: 10.1007/s11604-017-0617-z.

Reviewer 2 Report

The authors have made a vast review with a title of “A destruction model of the vascular and lymphatic systems in the emergence of psychiatric symptoms”. The needed work has clearly been significant and important. Although, I did find some minor/major concerns during my review. These concerns are listed down below. Page and line numbering was done according to the PDF that was provided. The more overall comments are listed first and more specific ones later:

  • Overall, there are plenty adequate references (and recent ones too) to support the authors’ claims. However, there are also claims that are without references, e.g. page 3, lines 111-112, page 4, lines 176-178, page 5, lines 237-239 or page 8, lines 386-388. I suggest that those and similar claims are addressed with appropriate references.

  • The manuscript language is not adequate enough for scientific publication. While reading the manuscript, I had to think the sentences over and over to really understand what authors have possibly meant. Additionally, there were several grammatical errors in the manuscript, that made the reading harder. I didn’t list these errors/hard sentences point-by-point, because of dense appearance of them. Thus, I would advice for language check for whole manuscript. This would also help the future article readers, especially those who are not experts on the field.

  • Moreover, in Introduction and Main text, there were sections, where authors could have been more concise and straightforward with their text, i.e. there were some more-or-less irrelevant sections. However, keeping in mind the nature of review articles, this more excessive writing style is somewhat acceptable, and thus I do not comment this further.

  • While reading the manuscript, I was very pleasantly surprised how well vascular, lymphatic and glymphatic systems were described and yet not so pleasantly surprised how thin connections were made with psychiatric diseases. I even started to ask myself, was the manuscript even describing about psychiatric symptoms/diseases, because there were not that many connections as the manuscript title had suggested. Because of this, the statement in the beginning of the conclusion was little surprising. Also, due to this, the conclusion chapter was the most informative when compared to original manuscript title and objective (page 2, lines 88-92). Thus, I would suggest authors to either emphasize the connections to psychiatric symptoms/diseases in the main text or modify the title of the manuscript.

  • The story of the manuscript felt like it was bouncing as I progressed. Maybe more concentrated sections/paragraphs with additional describing subheadings would help with it.  

  • The abbreviations were sometimes opened, sometimes not, e.g. first mention of CSF in page 3, line 100. Additionally, sometimes abbreviations were abbreviated even though, they were not used later, e.g. Huntington’s disease (HD), page 5, line 214-> there is no need to do such thing. Please modify all similar cases.

  • I do not think that it is appropriate to write abbreviations in the Abstract (page 1, line 29), and in Simple summary, i.e. there is no need to abbreviate CSF (page 1, line 11) and BBB (page 1, line 14).

  • Page 2, line 69, the text talks about BBB, but when looking the Figure 1, there is no mention of where BBB is in either in Figure or the Figure legend?

  • The figures itself are great, although the legends are quite long, but I think that cannot be helped. This is so that they are descriptive enough.

  • Some abbreviations/terms were written differently within the manuscript, e.g. Lyve-1 vs LYVE-1, SOX18 vs. Sox18 and VEGFC vs. VegfC. Please be congruent with the terms in whole manuscript.

  • Regarding to Glia-lymphatic system chapter, it would be appropriate to mention following articles that proved the system with humans: “Cerebrospinal fluid tracer efflux to parasagittal dura in humans” (https://www.nature.com/articles/s41467-019-14195-x) and “Glymphatics Visualization after Focused Ultrasound‐Induced Blood–Brain Barrier Opening in Humans” (https://onlinelibrary.wiley.com/doi/10.1002/ana.25604).

  • Furthermore, it has been shown, that brain lymphatics / glymphatics are not only driven through pulsations of arteries (cardiac), but additionally by respiratory pulsations and vasomotion (“Ultra-fast magnetic resonance encephalography of physiological brain activity - Glymphatic pulsation mechanisms?”, https://pubmed.ncbi.nlm.nih.gov/26690495/)

  • The AQP-4 hypothesis is one of the main mechanisms behind glymphatics. Thus, it would be appropriate to mention other possible factors contributing, such as “The perivascular astroglial sheath provides a complete covering of the brain microvessels: an electron microscopic 3D reconstruction”, https://onlinelibrary.wiley.com/doi/abs/10.1002/glia.20990.

  • The articles, “Disease-associated astrocytes in Alzheimer’s disease and aging”, https://www.nature.com/articles/s41593-020-0624-8, “Loss of Astrocytic Domain Organization in the Epileptic Brain”, https://www.jneurosci.org/content/28/13/3264.long and “Loss of perivascular aquaporin 4 may underlie deficient water and K+ homeostasis in the human epileptogenic hippocampus”, https://www.pnas.org/content/102/4/1193, could also give some more insights for role of the astrocytes and AQP-4 behind AD and epilepsy, and thus additionally others diseases?

  • All previous articles considered it might be interesting to discuss how the glymphatics is affected by differences of AQP-4/astrocytes and in addition how these and the physiological pulsations contributes to the glymphatics in psychiatric diseases, e.g. schizophrenia. This would be interesting, because there already are published articles on how physiological pulsations are contributing to glymphatics in brain diseases, e.g. “The variability of functional MRI brain signal increases in Alzheimer's disease at cardiorespiratory frequencies”, https://www.nature.com/articles/s41598-020-77984-1 and “Respiratory-related brain pulsations are increased in epilepsy—a two-centre functional MRI study”, https://academic.oup.com/braincomms/article/2/2/fcaa076/5854867

  • Page 9, lines 448-449, the opening of BBB could also be measured with DC-EEG, “Real-time monitoring of human blood-brain barrier disruption”, https://journals.plos.org/plosone/article?id=10.1371/journal.pone.0174072

  • Page 10, lines 458-460, the permeability of BBB naturally affects drug treatments, however, there are possibilities to improve this in the future “Noninvasive Ultrasonic Glymphatic Induction Enhances Intrathecal Drug Delivery”, https://www.biorxiv.org/content/10.1101/2020.10.21.348078v1.full. Although, this is not officially published yet, I thought it would be worth to mention to the authors.  

Author Response

Thank you so much for your very reasonable and constructive comments. I deeply appreciate your diligence and suggestions from your plenty of knowledge, that must cultivate our manuscript. Below, I give the point-by-point responses.

  • Overall, there are plenty adequate references (and recent ones too) to support the authors’ claims. However, there are also claims that are without references, e.g. page 3, lines 111-112, page 4, lines 176-178, page 5, lines 237-239 or page 8, lines 386-388. I suggest that those and similar claims are addressed with appropriate references.

Thank you for the comments. I appended the citations.

page 3, lines 111-112/new lines 120: [5](Aspelund et al., 2015)

page 4, lines 176-178/new lines 193: [40](Francois et al., 2008)

page 5, lines 237-239/new lines 245: [50,61]( Pekny et al., 2016; Pekny & Pekna, 2016)

page 8, lines 386-388/new lines 407, page 9: [122](Tay et al., 2017)

Plus, New page 12, new lines 537: [165](Perry et al., 1997)

New page 3, new lines 136: [27](Schulte-Merker et al., 2011)

, and else.

  • The manuscript language is not adequate enough for scientific publication. While reading the manuscript, I had to think the sentences over and over to really understand what authors have possibly meant. Additionally, there were several grammatical errors in the manuscript, that made the reading harder. I didn’t list these errors/hard sentences point-by-point, because of dense appearance of them. Thus, I would advice for language check for whole manuscript. This would also help the future article readers, especially those who are not experts on the field.

Thank you for the comments. I could understand why you had an impression of being under the borderline for scientific publication. The main reason is many errors od descriptions and less-conciseness, leading to less readability of this manuscript. It is my apology. Therefore, I tried the sentences and expressions mote straight and concise, hoping they would satisfy you. We checked all the sentences, and corrected errors and typos thoroughly as possible. All of them would make this manuscript reader-friendly and satisfy you, hopefully.

  • Moreover, in Introduction and Main text, there were sections, where authors could have been more concise and straightforward with their text, i.e. there were some more-or-less irrelevant sections. However, keeping in mind the nature of review articles, this more excessive writing style is somewhat acceptable, and thus I do not comment this further.

Thank you for your comment and impression of the manuscript. Well, three authors had described this initial version, but it appears not organized and not in a simple style. I just unified the style.

  • While reading the manuscript, I was very pleasantly surprised how well vascular, lymphatic and glymphatic systems were described and yet not so pleasantly surprised how thin connections were made with psychiatric diseases. I even started to ask myself, was the manuscript even describing about psychiatric symptoms/diseases, because there were not that many connections as the manuscript title had suggested. Because of this, the statement in the beginning of the conclusion was little surprising. Also, due to this, the conclusion chapter was the most informative when compared to original manuscript title and objective (page 2, lines 88-92). Thus, I would suggest authors to either emphasize the connections to psychiatric symptoms/diseases in the main text or modify the title of the manuscript.

I am also glad to hear from you like that. The current vascular system and aberrant immunity are getting very hot. And I agree with your comment indicating not well-described regarding “how thin connections were made with psychiatric diseases”.

We appended a new chapter indicating the link between the vascular, lymphatic systems, and the emergence of psychiatric disorders, to describe “how thin connections were made with psychiatric diseases”.

(new lines 571-, page 12)

  1. iv) Implications of the inflammation and immunity for the pathophysiology of psychiatric disorders

with additional citations.

  • The story of the manuscript felt like it was bouncing as I progressed. Maybe more concentrated sections/paragraphs with additional describing subheadings would help with it.  

Thank you for the comment. I totally agree with you. I appended the subheadings:

i) Vascular lymphatic system in the brain

i-1, Brain vasculature system

i-2, Characteristics of meningeal lymphatic system

i-3, Lymphatic vasculogenesis

ii) Glia-lymphatic system (Glymphatic system)

ii-1, Glymphatic system

ii-2, Astrocytic contribution and AQP-4 function

ii-3, Potential pathophysiology of the glymphatic system

iii) Infiltration of immune cells to brain across vasculature system

iii-1, Immune privilege

iii-2, Glial cell-characteristics of the brain and the regulation by immunity

iii-3, Infiltration of immune cells to brain parenchyma

iii-4, The structure and dysfunction of the BBB

iv) Implications of the inflammation and immunity for the pathophysiology of psychiatric disorders

  • The abbreviations were sometimes opened, sometimes not, e.g. first mention of CSF in page 3, line 100. Additionally, sometimes abbreviations were abbreviated even though, they were not used later, e.g. Huntington’s disease (HD), page 5, line 214-> there is no need to do such thing. Please modify all similar cases.

Thank you for your indication. I double-checked the abbreviations thoroughly and modify them all. I appreciate your kind comment. Change the abbreviations thoroughly, but I intentionally left several abbreviations (LIF, CCL2, EAAT2, PLP, Olig1-3, IBA1, LRP1, ZO-1, and ACE2) to indicate the protein names. AIDS and SLE are the prevailed disease names, and I left them as they are.

  • I do not think that it is appropriate to write abbreviations in the Abstract (page 1, line 29), and in Simple summary, i.e. there is no need to abbreviate CSF (page 1, line 11) and BBB (page 1, line 14).

Thank you for your indication as above. I removed both in the simple summary. And, in the abstract, the three terms CSF, ISF, and BBB are abbreviated.

  • Page 2, line 69, the text talks about BBB, but when looking the Figure 1, there is no mention of where BBB is in either in Figure or the Figure legend?

Thank you for the comment. That is another mistake. I amend the legend and drawing of the previous Figure 1. I clarified the BBB both in the drawing and legend.

New lines 645 (page 15 in new Figure2 legend):

This organized structure is the blood-brain barrier (BBB). Blood vessels are made up of endothelial cells (ECs) and mural cells on the abluminal surface of the endothelial-cell layer. Adjoined ECs are joined at tight and adhesion junctions. BBB is formed by endothelial cells, pericytes embedded in the capillary basement membrane, and astrocyte end-feet that enclose the capillary in a sheath.

  • The figures itself are great, although the legends are quite long, but I think that cannot be helped. This is so that they are descriptive enough.

I tried to make the Figure legends concise, transfer sentences to the main text, and gave other minor changes.

  • Some abbreviations/terms were written differently within the manuscript, e.g. Lyve-1 vs LYVE-1, SOX18 vs. Sox18 and VEGFC vs. VegfC. Please be congruent with the terms in whole manuscript.

That is my failure. Thank you for your kind indications. I unified and reorganized the style of mattered abbreviations, as pointed out in your previous comment.

  • Regarding to Glia-lymphatic system chapter, it would be appropriate to mention following articles that proved the system with humans: “Cerebrospinal fluid tracer efflux to parasagittal dura in humans” (https://www.nature.com/articles/s41467-019-14195-x) and “Glymphatics Visualization after Focused Ultrasound‐Induced Blood–Brain Barrier Opening in Humans” (https://onlinelibrary.wiley.com/doi/10.1002/ana.25604).

Thank you for your indication. I added the citation Meng et al., 2019 and relevant explanation to the main text:

New lines 300- (page 7 top):

A recent case report study showed the artificial opening of BBB using transcranial focused ultrasound in human brain, which proved the system in humans and the persistence of glymphatic efflux in AD and ALS patients [90](Meng et al., 2019).

Meng, Y.; Abrahao, A.; Heyn, C.C.; Bethune, A.J.; Huang, Y.; Pople, C.B.; Aubert, I.; Hamani, C.; Zinman, L.; Hynynen, K.; Black, S.E.; Lipsman, N. Glymphatics Visualization after Focused Ultrasound-Induced Blood-Brain Barrier Opening in Humans. Ann Neurol. 2019, 86, 975–980. doi: 10.1002/ana.25604.

  • Furthermore, it has been shown, that brain lymphatics / glymphatics are not only driven through pulsations of arteries (cardiac), but additionally by respiratory pulsations and vasomotion (“Ultra-fast magnetic resonance encephalography of physiological brain activity - Glymphatic pulsation mechanisms?”, https://pubmed.ncbi.nlm.nih.gov/26690495/)

Thank you for your comment. We had not known the result of Kiviniemi et al. (2016), while this paper seems including erroneous values and confuses me. I appended your suggesting citation and modified relevant sentences.

New lines 110- (page 3):

According to Kiviniemi et al. (2016) [22], other than cardiac pulsation (1.08Hz), they proved additional two types of physiological mechanisms affecting CSF/ISF pulsations: respiratory pulsation (0.37 Hz), and very low frequency vasomotor pulsations (at 0.01–0.027 Hz and 0.027–0.073 Hz) [22]( Kiviniemi et al., 2016).

Kiviniemi, V.; Wang, X.; Korhonen, V.; Keinänen, T.; Tuovinen, T.; Autio, J.; LeVan, P.; Keilholz, S.; Zang, Y-F.; Hennig, J.; Nedergaard, M. Ultra-fast magnetic resonance encephalography of physiological brain activity - Glymphatic pulsation mechanisms? J Cereb Blood Flow Metab. 2016, 36, 1033–1045. doi: 10.1177/0271678X15622047.

  • The AQP-4 hypothesis is one of the main mechanisms behind glymphatics. Thus, it would be appropriate to mention other possible factors contributing, such as “The perivascular astroglial sheath provides a complete covering of the brain microvessels: an electron microscopic 3D reconstruction”, https://onlinelibrary.wiley.com/doi/abs/10.1002/glia.20990.

Thank you for your comment. I understand it. It was critical. And, sorry for my immature writing. In the initial version, we ended up casting a doubt on the AQP4 hypothesis (Haj-Yasein et al., 2011 [11]), but the supportive result was shown in Iliff et al. (2012)[2], as well as Mathiisen et al. (2010 Glia) as your suggestion. Therefore, the previous description did not show the current schema correctly. I’ve changed the sentences as below:

New lines 280- (page 6):

A following study by Iliff et al. (2012)[2] demonstrated that the fluorescence of small-, but not large-, molecular weight interstitial tracer less spread in the AQP4-null mouse brains than control, suggesting the AQP4-dependent interstitial solute and fluid clearance in the brain. The perivascular astroglial sheath is completely continuous and covering the capillary surface without gaps, and therefore, the AQP-4 would play rate limiting role of astrocytic endfeet [79](Mathiisen et al., 2010).

Mathiisen, T.M.; Lehre, K.P.; Danbolt, N.C.; Ottersen, O.P. The perivascular astroglial sheath provides a complete covering of the brain microvessels: an electron microscopic 3D reconstruction. Glia 2010, 58, 1094–1103. doi: 10.1002/glia.20990.

  • The articles, “Disease-associated astrocytes in Alzheimer’s disease and aging”, https://www.nature.com/articles/s41593-020-0624-8, “Loss of Astrocytic Domain Organization in the Epileptic Brain”, https://www.jneurosci.org/content/28/13/3264.long and “Loss of perivascular aquaporin 4 may underlie deficient water and K+ homeostasis in the human epileptogenic hippocampus”, https://www.pnas.org/content/102/4/1193, could also give some more insights for role of the astrocytes and AQP-4 behind AD and epilepsy, and thus additionally others diseases?

Thank you for your insightful comment. I changed the sentences following to your indications as below:

new lines 313- (page 7):

Other evidence for the roles of the astrocytes and AQP-4 behind AD and epilepsy may also give more insights [96-98] (Eid et al., 2005; Oberheim et al., 2008; Habib et al., 2020). In the epileptic brain, the extracellular K+-buffering, AQP-4 localization, and domain organization of astrocyte processes are disrupted in mouse models and human patients [96,97] (Eid et al., 2005; Oberheim et al., 2008). Loss of perivascular AQP4 might be involved in the pathogenesis of mesial temporal lobe epilepsy (MTLE). The density of AQP4 along the perivascular membrane of astrocytes endfeet was reduced by 44% in hippocampal CA1 of MTLE compared to non-MTLE in human samples, suggesting leading to an impaired water and K+-homeostasis in MTLE parenchyma [96] (Eid et al., 2005). In mouse models of epilepsy, cortical astrocytes show the morphological changes and increase in overlap of their processes [97] (Oberheim et al., 2008).

Habib, N.; McCabe, C.; Medina, S.; Varshavsky, M.; Kitsberg, D.; Szternfeld, R.D.; Green, G.; Dionne, D.; Nguyen, L.; Marshall, J.L.; Chen, F.; Zhang, F.; Kaplan, T.; Regev, A.; Schwartz, M. Disease-associated astrocytes in Alzheimer's disease and aging. Nat Neurosci. 2020, 23, 701–706. doi: 10.1038/s41593-020-0624-8.

Oberheim, N.A.; Tian, G-F.; Han, X.; Peng, W.; Takano, T.; Ransom, B.; Nedergaard, M. Loss of astrocytic domain organization in the epileptic brain. J Neurosci. 2008, 28, 3264–3276. doi: 10.1523/JNEUROSCI.4980-07.2008.

Eid, T.; Lee, T-S.W.; Thomas, M.J.; Amiry-Moghaddam, M.; Bjørnsen, L.P.; Spencer, D.D.; Agre, P.; Ottersen, O.P.; de Lanerolle, N.C. Loss of perivascular aquaporin 4 may underlie deficient water and K+ homeostasis in the human epileptogenic hippocampus. Proc Natl Acad Sci U S A. 2005, 102, 1193–1198. doi: 10.1073/pnas.0409308102.

  • All previous articles considered it might be interesting to discuss how the glymphatics is affected by differences of AQP-4/astrocytes and in addition how these and the physiological pulsations contributes to the glymphatics in psychiatric diseases, e.g. schizophrenia. This would be interesting, because there already are published articles on how physiological pulsations are contributing to glymphatics in brain diseases, e.g. “The variability of functional MRI brain signal increases in Alzheimer's disease at cardiorespiratory frequencies”, https://www.nature.com/articles/s41598-020-77984-1 and “Respiratory-related brain pulsations are increased in epilepsy—a two-centre functional MRI study”, https://academic.oup.com/braincomms/article/2/2/fcaa076/5854867

Thank you for your suggestion of interesting papers. Brain BOLD signal variability in patients with AD, mainly due to the cardiovascular brain pulsations, is quite interesting notion (Tuovinen et al., 2020). I expanded the sentences following to your indications as below:

New lines 268- (page 6):

Interestingly, a recent study reported a variability of the brain BOLD (blood oxygen level-dependent) signal in patients with AD, mainly due to the cardiovascular pulsations in the brain parenchyma [77](Tuovinen et al., 2020). The authors detected neither differences in the average cardiorespiratory rates nor the blood pressure between the control and AD groups [77](Tuovinen et al., 2020). Thus, the impairment of the brain vasculature system could be used as the biomarker for detecting AD patients.

Tuovinen, T.; Kananen, J.; Rajna, Z.; Lieslehto, J.; Korhonen, V.; Rytty, R.; Mattila, H.; Huotari, N.; Raitamaa, L.; Helakari, H.; Elseoud, A.A.; Krüger, J.; LeVan, P.; Tervonen, O.; Hennig, J.; Remes, A.M.; Nedergaard, M.; Kiviniemi, V. The variability of functional MRI brain signal increases in Alzheimer's disease at cardiorespiratory frequencies. Sci Rep. 2020, 10, 21559. doi: 10.1038/s41598-020-77984-1.

New lines 321- (page 7):

Therefore, mechanisms driving the CSF homeostasis are thought to be altered in epilepsy. Indeed, in contrast to AD patients, respiratory-related brain pulsations are increased in epilepsy patients without changes in physiological cardiorespiratory rates [99](Kananen et al., 2020). The most affected region of the brain was the upper brain stem respiratory pneumotaxic centre, midbrain and temporal lobes, including amygdalae, hippocampi, pallida and putamina, almost of all of those are close to the cavernous sinus in epilepsy patients [99](Kananen et al., 2020).

Kananen, J.; Helakari, H.; Korhonen, V.; Huotari, N.; Järvelä, M.; Raitamaa, L.; Raatikainen, V.; Rajna, Z.; Tuovinen, T.; Nedergaard, M.; Jacobs, J.; LeVan, P.; Ansakorpi, H.; Kiviniemi, V. Respiratory-related brain pulsations are increased in epilepsy-a two-centre functional MRI study. Brain Commun. 2020, 2, fcaa076. doi: 10.1093/braincomms/fcaa076.

  • Page 9, lines 448-449, the opening of BBB could also be measured with DC-EEG, “Real-time monitoring of human blood-brain barrier disruption”, https://journals.plos.org/plosone/article?id=10.1371/journal.pone.0174072

Thank you for the additional paper.

new lines 464- (page 10):

Current advanced MRI scanning and the direct-current electroencephalography enable the clinical diagnostic test to detect such BBB breakdowns [83,84,141](Montagne et al., 2015; Nation et al., 2019; Kiviniemi et al., 2017).

Kiviniemi, V.; Korhonen, V.; Kortelainen, J.; Rytky, S.; Keinänen, T.; Tuovinen, T.; Isokangas, M.; Sonkajärvi, E.; Siniluoto, T.; Nikkinen, J.; Alahuhta, S.; Tervonen, O.; Turpeenniemi-Hujanen, T.; Myllylä, T.; Kuittinen, O.; Voipio, J. Real-time monitoring of human blood-brain barrier disruption. PLoS One 2017, 12, e0174072. doi: 10.1371/journal.pone.0174072.

  • Page 10, lines 458-460, the permeability of BBB naturally affects drug treatments, however, there are possibilities to improve this in the future “Noninvasive Ultrasonic Glymphatic Induction Enhances Intrathecal Drug Delivery”, https://www.biorxiv.org/content/10.1101/2020.10.21.348078v1.full. Although, this is not officially published yet, I thought it would be worth to mention to the authors.  

Thank you for the additional paper. Exactly. The BBB opening technique appears increase the efficacy of drug delivery, and this gives a kind of hopes for future.

New lines 479- (page 10):

It would be worth mentioning that such a poor parenchymal uptake of agents from the CSF via BBB and glymphatic fluid transport are now overcome by the transcranial ultrasound [146,147](Aryal et al., 2020; Haumann et al., 2020). Ultrasound protocol significantly improves the brain parenchymal uptake of drugs and antibodies (i.e., by around 4 times in case of 1 kDa MRI tracer) via intrathecal administration [146](Aryal et al., 2020).

Aryal, M.; Zhou, Q.; Rosenthal, E.L.; Airan, R.D. Noninvasive Ultrasonic Glymphatic Induction Enhances Intrathecal Drug Delivery. bioRxiv 2020. doi: 10.1101/2020.10.21.348078.

Haumann, R.; Videira, J.C.; Kaspers, G.J.L.; van Vuurden, D.G.; Hulleman, E. Overview of Current Drug Delivery Methods Across the Blood-Brain Barrier for the Treatment of Primary Brain Tumors. CNS Drugs 2020, 34, 1121–1131. doi: 10.1007/s40263-020-00766-w.

Reviewer 3 Report

This is a very timely review about an important topic which is sometimes not well defined. The following can improve the reader’s understanding before publication.

  1. The CNS lymphatic system is indeed relatively (re)new and there are some confusions regarding its anatomical features. It would be helpful if a better description including general schematics of the ALL CNS lymphatic system in comparison to the CNS vasculature system will appear in the manuscript. Specifically, please elaborate regarding the anatomy, location in CNS, size and in general ways to specifically differentiate between the blood vessels and the lymphatic vessels.
  2. The interaction between lymphatic vessels and blood vessels is not clear (Schema 1). It is said that they are “associated”. How exactly? In scheme 1 it is not clear how the lymphatic vessels are connected to the veins.
  3. A paragraph on BCSFB is missing since there is a lot of immune cells activity in the choroid plexus that needed to be addressed in comparison to BBB and lymphatic system.
  4. References appear both in numbers and names. Is that according to the journal style?
  5. There are several grammar mistakes throughout the manuscript
  6. If possible to give real pictures of the lymphatic system rather than schematic it will be very insightful
  7. Lyve-1 is mentioned as a (endothelial?) lymphatic marker. Please provide more markers and elaborate on ways to differentiate between EC of BBB, larger blood vessels and lymphatic vessels. Is there a way to isolate lymphatic vessels from animals and human brains? For example, when investigators isolate blood vessels from brains, do they isolate also lymphatic vessels and are not aware of that? Is it possible to differentiate between aqp4 of astrocytes related to BBB to those of LV? What are these “zipper-like junctions” made of (tight junctions? The same as in the BBB?)?
  8. “dopaminergic neurons” instead of “dopamine neurons”
  9. Is all the CNS lymphatic system is considered “glymphatic”? it is not completely clear from the text. Again, an anatomical picture of all the lymphatic picture in all the CNS will be helpful.
  10. Line 307- can you provide the original reference?
  11. Lines 407-431- it is not clear if this description is only for peripheral system or also for the CNS
  12. Line 439- is the “glymphatic system” appear in fig.2?
  13. Line 446: what do you mean by “BBB rupture also looks like a type of EAE” ?
  14. Line 458: please verify that “98% “ referes to failures in clinical trials and not to the amount of molecules that don’t cross the BBB.
  15. Lines 520-521: please provide references for “…without the BBB breaking down”
  16. Line 590: “morphology correlated”: please clearly define this morphology correlation and connections

Author Response

Thank you so much for your comment. I deeply appreciate them and give the point-by-point responses. Your diligence and thoughtful comments encouraged us.

  1. The CNS lymphatic system is indeed relatively (re)new and there are some confusions regarding its anatomical features. It would be helpful if a better description including general schematics of the ALL CNS lymphatic system in comparison to the CNS vasculature system will appear in the manuscript. Specifically, please elaborate regarding the anatomy, location in CNS, size and in general ways to specifically differentiate between the blood vessels and the lymphatic vessels.

Thank you for your indications. I separate your comments to three points.

(1) “It would be helpful if a better description including general schematics of the ALL CNS lymphatic system”

We append the new Figure 1: Anatomy of vasculature systems in human and mouse brains. There, we provide the whole lymphatic system in the brains, within our current knowledge, and their relationship to other organs: blood vessels and ventricles.

(2) “Anatomy, location in CNS, size and in general ways”

[Anatomy, location, & size]

According to Aspelund et al., 2015:

(Mouse) in the rostral rhinal vein, superior sagittal sinus, transverse vein, sigmoid sinus, retroglenoid vein, middle meningeal artery, and pterygopalatine artery. Plus, cribriform plate as the initiation of the nasal lymphatics.

Size: 10-40 μm diameter.

According to Absinta et al., 2017:

(Human) Superior sagittal sinus, straight sinus, cingulate gyrus, and middle meningeal artery. Plus, cribriform plate.

Size: from 30 μm to 300 μm in diameter

(Marmoset) Superior sagittal sinus, straight sinus

(3) Differences “between the blood vessels and the lymphatic vessels.”

[Marker proteins]

Lymph marker: LYVE-1, Prox-1, PDPN (Podoplanin), VEGFR3, CCL21

Blood vessel marker: CD31 (PECAM-1: platelet endothelial cell adhesion molecule)

(Aspelund et al., 2015; Louveau et al., 2015; Louveau et al., 2018)

This information above (2) & (3) is described briefly in the main text and Figure1 legend.

Though it is not cited in the manuscript, the study below may assist your knowledge.

Jeffrey J Lochhead, Thomas P Davis. Perivascular and Perineural Pathways Involved in Brain Delivery and Distribution of Drugs after Intranasal Administration. Pharmaceutics. 2019 Nov 12;11(11):598. doi: 10.3390/pharmaceutics11110598.

New line 624- (page 14, Figure 1 legend):

According to Absinta et al. (2017)[8], in human brain, the brain lymphatic vessels (LVs) are found in the superior sagittal sinus, straight sinus, cingulate gyrus, and middle meningeal artery. Meningeal LVs bilaterally run in parallel to the veins and sinuses. Additionally, LVs are found in cribriform plate as the initiation of the nasal lymphatics. The size of vessels is from 30 μm to 300 μm in diameter, depending on the location. In the marmoset brain, LVs are found in the superior sagittal sinus, and straight sinus [8](Absinta et al., 2017). According to Aspelund et al. (2015)[5], in mouse brain, LVs are found in the rostral rhinal vein, superior sagittal sinus, transverse vein, sigmoid sinus, retroglenoid vein, middle meningeal artery, and pterygopalatine artery, in addition to the cribriform plate with 10-40 μm diameter [5](Aspelund et al., 2015).

New line 156- (page 4):

Briefly, researchers use the marker proteins of the brain lymphatic vessels: LYVE-1, prospero homeobox protein 1 (Prox-1), PDPN (Podoplanin), VEGFR3, and chemokine (C-C motif) ligand 21 (CCL21). It is distinguishable from the expression of a blood vessel marker protein: CD31 (PECAM-1, platelet endothelial cell adhesion molecule-1) [4,5,21] (Aspelund et al., 2015; Louveau et al., 2015; Louveau et al., 2018).

2. The interaction between lymphatic vessels and blood vessels is not clear (Schema 1). It is said that they are “associated”. How exactly? In scheme 1 it is not clear how the lymphatic vessels are connected to the veins.

Lymphatic vessels are known to differentiate from veins.

Meningeal LVs bilaterally run in parallel to the veins and sinuses.

In mouse, Lymphatic vessel diameter is around 20-25 (μm) (Louveau et al., 2015) apart from blood vessels by 10-20 μm (venules) or 50 μm (sinusal lumen) (Aspelund et al., 2015; Louveau et al., 2015; Brezovakova & Jadhav, 2020). In human, LV locate around the sinus and blood vessels (Absinta et al., 2017) as aforementioned.

The information of distance from the vein and sinus are not texted in the manuscript to avoid further complexity.

Brezovakova, V.; Jadhav, S. Identification of Lyve-1 positive macrophages as resident cells in meninges of rats. J Comp Neurol. 2020, 528, 2021–2032. doi:10.1002/cne.24870.

Thank you for your questions.

3. A paragraph on BCSFB (Brain-CSF barrier) is missing since there is a lot of immune cells activity in the choroid plexus that needed to be addressed in comparison to BBB and lymphatic system.

Although the BCSFB is an important mechanism for the CSF production in the choroid plexus, we had thought it is needless. In this revised manuscript, we appended the description as below. I appreciate your suggestion.

New line 103 (Page 3):

Ependymal cells also have a fluid–brain barrier function between ventricles and parenchyma, which is called the blood-cerebrospinal fluid barrier (BCSFB). Ependymal cells form the choroid plexus, and it has a high water and ion permeability. The choroid plexus absorbs water and ions from blood much more efficiently than the BBB and generates CSF in the ventricles and subarachnoid spaces (Figure 1).

New line 113 (Page 3):

Ciliary motility of ependymal cells is also known involved in the efficient and continuous maintenance of CSF flow near the ventricular surface [23](Sawamoto et al., 2006).

Sawamoto, K.; Wichterle, H.; Gonzalez-Perez, O.; Cholfin, J.A.; Yamada, M.; Spassky, N.; Murcia, N.S.; Garcia-Verdugo, J.M.; Marin, O.; Rubenstein, J.L.R.; Tessier-Lavigne, M.; Okano, H.; Alvarez-Buylla, A. New neurons follow the flow of cerebrospinal fluid in the adult brain. Science 2006, 311, 629-632. doi: 10.1126/science.1119133.

4. References appear both in numbers and names. Is that according to the journal style?

Thank you for the comment.

In the initial version, we intentionally left the (author + et al., Years) to follow the citations easily. Here, in this revised version, they all are modified.

5. There are several grammar mistakes throughout the manuscript

Thank you for the indication. We enthusiastically refined the sentences and errors.

6. If possible to give real pictures of the lymphatic system rather than schematic it will be very insightful

Thank you for the comment. However, there are no adequate pictures of the lymphatic system to provide at this time. I appreciate your kind suggestion, but we cannot respond to this comment. Very sorry.

7. Lyve-1 is mentioned as a (endothelial?) lymphatic marker. Please provide more markers and elaborate on ways to differentiate between EC of BBB, larger blood vessels and lymphatic vessels. Is there a way to isolate lymphatic vessels from animals and human brains? For example, when investigators isolate blood vessels from brains, do they isolate also lymphatic vessels and are not aware of that? Is it possible to differentiate between aqp4 of astrocytes related to BBB to those of LV? What are these “zipper-like junctions” made of (tight junctions? The same as in the BBB?)?

Thank you for the comment. I gave responses to each question:

/ Lyve-1 is mentioned as a (endothelial?) lymphatic marker. Please provide more markers and elaborate on ways to differentiate between EC of BBB, larger blood vessels and lymphatic vessels.

LYVE1 is a membrane glycoprotein, which has a function of the receptor for both soluble and immobilized hyaluronan. LYVE-1 is a cell surface receptor on lymphatic endothelial cells. The markers are introduced as a response to your previous comment. Again, in general, the vasculatures are CD31-positive, and lymphatic vessels are LYVE-1 and PROX-1 positive. However, we know there are LYVE-1 positive venules in brain parenchyma. BBB exists in all the area of the brain if the capillary and astrocytes exist. ECs of choroid plexus (BCSFB region) are also LYVE-1 positive.

/Is there a way to isolate lymphatic vessels from animals and human brains?

It is possible to isolate the meningeal lymphatic vessels, and therefore, it may be able to establish a culture system in both from animal and human samples. In case of human samples, you can obtain biopsy during the surgical operation. Since the mechanism of maturation of meningeal lymphatic vessels is unknown, it is utterly important to establish such an experimental system, we would like to challenge it if a chance should come.

/For example, when investigators isolate blood vessels from brains, do they isolate also lymphatic vessels and are not aware of that?

As mentioned, it could be possible. Please note that the lymphatic vessels develop specifically from the veins. Tammela and Alitalo (2010, Cell)[37] and Risau (1997, Nature)[35] would help your knowledge.

Tammela T, Alitalo K. Lymphangiogenesis: Molecular mechanisms and future promise. Cell. 2010 Feb 19;140(4):460-76. https://www.ncbi.nlm.nih.gov/pubmed/20178740

Risau, W. Mechanisms of angiogenesis. Nature 1997, 386, 671–674. doi: 10.1038/386671a0.

/Is it possible to differentiate between aqp4 of astrocytes related to BBB to those of LV?

I have no idea. Astrocyte and the brain lymphatic vessels are developmentally different. LV endothelium is of mesodermal origin, and astrocytes are derived from neural stem cells and neural crest (ectodermal origin). Our writing might lead you confused, but please note that, as far as we know, no studies have examined the binding of astrocytes to lymphatic vessels.

/What are these “zipper-like junctions” made of (tight junctions? The same as in the BBB?)?

The initial lymphatics have distinctive, discontinuous buttons in endothelium, while the collecting lymphatics have continuous zippers. Both types of junctions consist of proteins typical of adherens junctions (AJs) and tight junctions (TJs). “Zipper-like junctions” are made of AJs (VE-cadherin, and many other molecules) and TJs (Occludin, Claudins, and Junction Adhesion Molecules (JAM)).

Endothelial cells of capillaries are physically connected with each other through the TJ, adherens AJ, and gap junction proteins. Endothelial TJ and AJ proteins themselves constitute the physical barrier of the BBB.

I could not understand how much molecular subfamilies are shared or independent in endothelial cells of venules and lymphatic vessels.

We modified the following sentences (new lines 127-132, page 3):

In general, in the body, the lymph is first drained into the initial lymphatic vessels that have both lymphatic capillaries, composed of lymphatic endothelial cells (LECs). The initial lymphatics have distinctive, discontinuous buttons in endothelium, while the collecting vessels are covered with a continuous basement membrane and smooth muscle cells (SMCs). Endothelial cells in collecting vessels is an elongated shape, and they are connected by continuous zipper-like junctions [27](Schulte-Merker et al., 2011). Both types of junction consist of proteins typical of adherens junctions (AJs) and tight junctions (TJs).

8. “dopaminergic neurons” instead of “dopamine neurons”

We changed the “dopamine neurons” to “dopaminergic neurons.”

9. Is all the CNS lymphatic system is considered “glymphatic”? it is not completely clear from the text. Again, an anatomical picture of all the lymphatic picture in all the CNS will be helpful.

“Glymphatic” identity is defined mainly by the astrocyte endfeet and AQP-4 expression. I don’t know the specific brain regions without astrocytes. But, the blood–cerebrospinal fluid barrier (BCSFB) of the choroid plexus and pituitary gland do not have the BBB, and so would be the glymphatic system. Sorry. I cannot answer the comment but hope it might satisfy you.

10. Line 307- can you provide the original reference?

There are no original references. We describe this part by summarizing my Japanese textbook of immunity, [100](Niederkorn, 2006) and Medawar [101](Medawar, 1948) very very concisely.

11. Lines 407-431- it is not clear if this description is only for peripheral system or also for the CNS

Thank you so much for your kind comment. I agree with you. The molecular mechanism of the immune cell infiltration into the brain parenchyma is still unclear or under debate. In this paragraph (Lines 407-431/new line 424-447), because of the lack in sufficient of studies in the CNS, we mainly discuss in the peripheral tissues. We gave minor changes in these sentences, and those should make the sentences clear.

12. Line 439- is the “glymphatic system” appear in fig.2?

Line 439-/new line 454- (page 10):

Yes. In the condition of aberrant immunity like as neurodegenerative diseases, the breakdown of the CNS vasculatures occurs. The glymphatic system is a pathway for the brain perfusion by CSF/interstitial fluid (ISF) for waste clearance (orange and blue gradient arrows in Figure 2). Therefore, the breakdown of the perivascular system includes the disruption of astrocyte endofeet, i.e., glympatic system, partially.

We consider it is not necessary to amend the figure. Thank you for your closer look. If this does not satisfy you, please kindly tell us in the next revision comment.

13. Line 446: what do you mean by “BBB rupture also looks like a type of EAE” ?

Line 446-/new line 462- (page 10):

Thanks. “BBB rupture also looks like a type of EAE [15](Sweeney et al., 2019),” is problematic. We changed the sentence as follows:

As inferred from the mouse EAE model, the CNS inflammation and disruption of the barrier structure can occur in BBB breakdown in humans [15](Sweeney et al., 2019).

14. Line 458: please verify that “98% “ referes to failures in clinical trials and not to the amount of molecules that don’t cross the BBB.

Line 458-/new line 475- (page 10)

Thank you for the comment. I amended the explanation as follows:

It has been noted that 98% of small-molecule drugs fail in clinical trials due to less BBB permeability. Only limited opiates, anxiolytics (e.g., benzodiazepines), antipsychotics (e.g. chlorpromazine), and anti-dementia drugs (e.g., donepezil, memantine, and tacrine) can cross the BBB; thus the efficacy of the drug treatment is limited [143, 145](Pardridge 2007; Caprifico et al., 2020).

Caprifico, A.E.; Foot, P.J.S.; Polycarpou, E.; Calabrese, G. Overcoming the Blood-Brain Barrier: Functionalised Chitosan Nanocarriers. Pharmaceutics 2020, 12, 1013. doi: 10.3390/pharmaceutics12111013.

15. Lines 520-521: please provide references for “…without the BBB breaking down”

We appended the reference. Thanks.

New line 537- (page 11):

without the BBB breaking down [165](Perry et al., 1997)

16. Line 590: “morphology correlated”: please clearly define this morphology correlation and connections

Line 590-/new line 663- (page 16):

The capillaries and basal meningeal lymphatic vessels are morphologically correlated; the capillaries have a peripheral morphology against the large blood vessels, while the basal meningeal lymphatic vessels also have a terminal/initial morphology against the large lymph vessels.

Thank you so much, again, for your all comment. While I did not have the insight from the view-point of development and molecular neuroscience, but your insightful comment critically improved our manuscript.

Round 2

Reviewer 2 Report

Thanks for the authors for revised manuscript. My previous points were mostly addressed. The structure got better and additionally the relevance in psychiatric diseases were addressed better, that were my main points before. Additionally, the new first figure was great.  

The language got slightly better, unfortunately I must still say that I suggest that the manuscript should go through language check. I think that there were verbose expressions, little "clumsy" structures and sentences in the manuscript. It is readable for sure, but if checked, the text could be more scientific, concise, understandable and thus more impactful. However, I may be alone with my opinion, so therefore, I think it is up to the Editor to choose how to proceed. 

That being said, I am satisfied with these modifications and manuscript should be published, if the language is okay for the Editor.